# Predictive Querying for Autoregressive Neural Sequence Models

**Alex Boyd**[*1]    **Sam Showalter**[*2]    **Stephan Mandt**[1,2]    **Padhraic Smyth**[1,2]
[1]Department of Statistics    [2]Department of Computer Science
University of California, Irvine
{alexjb,showalte,mandt,p.smyth}@uci.edu

## Abstract

In reasoning about sequential events it is natural to pose probabilistic queries such as "when will event A occur next" or "what is the probability of A occurring before B," with applications in areas such as user modeling, language models, medicine, and finance. These types of queries are complex to answer compared to next-event prediction, particularly for neural autoregressive models such as recurrent neural networks and transformers. This is in part due to the fact that future querying involves marginalization over large path spaces, which is not straightforward to do efficiently in such models. In this paper we introduce a general typology for predictive queries in neural autoregressive sequence models and show that such queries can be systematically represented by sets of elementary building blocks. We leverage this typology to develop new query estimation methods based on beam search, importance sampling, and hybrids. Across four large-scale sequence datasets from different application domains, as well as for the GPT-2 language model, we demonstrate the ability to make query answering tractable for arbitrary queries in exponentially-large predictive path-spaces, and find clear differences in cost-accuracy tradeoffs between search and sampling methods.

## 1   Introduction

One of the major successes in machine learning in recent years has been the development of neural sequence models for categorical sequences, particularly in natural language applications but also in other areas such as automatic code generation and program synthesis [Shin et al., 2019, Chen et al., 2021], computer security [Brown et al., 2018], recommender systems [Wu et al., 2017], genomics [Shin et al., 2021, Amin et al., 2021], and survival analysis [Lee et al., 2019]. Many of the models (although not all) rely on autoregressive training and prediction, allowing for the sequential generation of sequence completions in a recursive manner conditioned on sequence history.

A natural question in this context is how to compute answers to predictive queries that go beyond traditional one-step-ahead predictions. Examples of such queries are "how likely is event $A$ to occur before event $B$?" and "how likely is event $C$ to occur (once or more) within the next $K$ steps of the sequence?" These types of queries are very natural across a wide variety of application contexts, for example, the probability that an individual will finish speaking or writing a sentence within the next $K$ words, or that a user will use one app before another. See Fig. 1 and Appendix G for examples.

In this paper we develop a general framework for answering such predictive queries in the context of autoregressive (AR) neural sequence models. This amounts to computing conditional probabilities of propositional statements about future events, conditioned on the history of the sequence as summarized by the current hidden state representation. We focus in particular on how to perform near

---

[*]Authors contributed equally

36th Conference on Neural Information Processing Systems (NeurIPS 2022).

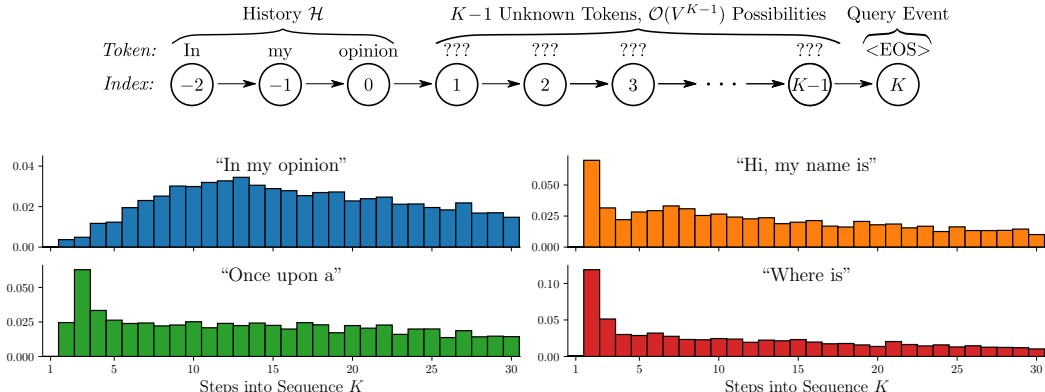

Figure 1: (top) Illustration of a query for the probability of a given sentence "In my opinion..." ending in $K$ steps. (bottom) GPT-2 query estimates across 4 prefixes with $V = 50,257, K \leq 30$. Importance sampling query estimates maintain a 6x reduction in variance relative to naive model sampling for the same computation budget. Open-ended prefixes (top-left) generally possess longer-tailed distributions relative to simple prefixes. Additional details provided in Sections 4, 5, and Appendix H.5.

real-time computation of such queries, motivated by use-cases such as answering human-generated queries and utilizing query estimates within the optimization loop of training a model. Somewhat surprisingly, although there has been extensive prior work on multivariate probabilistic querying in areas such as graphical models and database querying, as well as for restricted types of queries for traditional sequence models such as Markov models, querying for neural sequence models appears to be unexplored. One possible reason is that the problem is computationally intractable in the general case (as we discuss later in Section 3), typically scaling as $\mathcal{O}\big(V^{K-1}\big)$ or worse for predictions $K$-steps ahead, given a sequence model with a $V$-ary alphabet (e.g. compared to $\mathcal{O}\big(KV^2\big)$ for Markov chains).

Our contributions are three-fold:

1. We introduce and develop the problem of predictive querying in neural sequence models by reducing complex queries to building blocks im the form of elementary queries. These elementary queries define restricted path spaces over future event sequences.

2. We show that the underlying autoregressive model can always be constrained to the restricted path space satisfying a given query. This gives rise to a novel proposal distribution that can be used for importance sampling, beam search, or a new hybrid approach.

3. We evaluate these methods across three user behavior datasets and two language datasets. While all three methods significantly improve over naive forward simulation, the hybrid approach further improves over importance sampling and beam search. We furthermore explore how the performance of all methods relates to the model entropy.

Code for this work is available at https://github.com/ajboyd2/prob_seq_queries.

## 2 Related Work

Research on efficient computation of probabilistic queries has a long history in machine learning and AI, going back to work on exact inference in multivariate graphical models [Pearl, 1988, Koller and Friedman, 2009]. Queries in this context are typically of two types. The first are *conditional probability queries*, which are the focus of our attention here: computing probabilities defined for a subset $X$ of variables of interest, conditioned on a second subset $Y = y$ of observed variable values, and marginalizing over the set $Z$ of all other variables. The second type of queries can broadly be referred to as *assignment queries*, seeking the most likely (highest conditional probability) assignment of values $x$ for $X$, again conditioned on $Y = y$ and marginalizing over the set $Z$. Assignment queries are also referred to as most probable explanation (MPE) queries, or as maximum a posteriori (MAP) queries when $Z$ is the empty set [Koller et al., 2007].

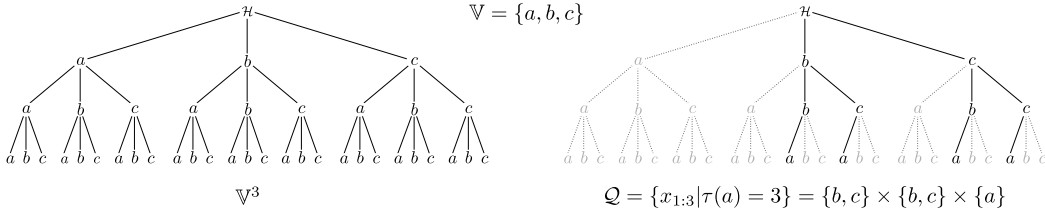

Figure 2: (left) Tree diagram of the complete sequence space for a vocabulary $\mathbb{V} = \{a, b, c\}$ and the corresponding query space $\mathcal{Q}$ (right) for when *the first appearance of $a$* occurs on the third step (i.e., $\tau(a) = 3$), defined as the set product of restricted domains listed below the figure.

For models that can be characterized with sparse Markov dependence structure, there is a significant body of work on efficient inference algorithms that can leverage such structure [Koller and Friedman, 2009], in particular for sequential models where recursive computation can be effectively leveraged [Bilmes, 2010]. However, autoregressive neural sequence models are inherently non-Markov since the real-valued current hidden state is a function of the entire history of the sequence. Each hidden state vector induces a tree containing $V^K$ unique future trajectories with state-dependent probabilities for each path of length $K$. Techniques such as dynamic programming (used effectively in Markov-structured sequence models) are not applicable in this context, and both assignment queries and conditional probability queries are NP-hard in general Chen et al. [2018].

For assignment-type queries there has been considerable work in natural language processing with neural sequence models, particularly for the MAP problem of generating high-quality/high-probability sequences conditioned on sequence history or other conditioning information. A variety of heuristic decoding methods have been developed and found to be useful in practice, including beam search [Sutskever et al., 2014], best-first search [Xu and Durrett, 2021], sampling methods [Holtzman et al., 2019], and hybrid variants [Shaham and Levy, 2022]. However, for conditional probability queries with neural sequence models (the focus of this paper), there has been no prior work in general on this problem to our knowledge. While decoding techniques such as beam search can also be useful in the context of conditional probability queries, as we will see later in Section 4, such techniques have significant limitations in this context, since by definition they produce lower-bounds on the probabilities of interest and, hence, are biased estimators.

## 3 Probabilistic Queries

**Notation** Let $X_{1:N} := [X_1, X_2, \ldots, X_N]$ be a sequence of random variables with arbitrary length $N$. Additionally, let $x_{1:N} := [x_1, x_2, \ldots, x_N]$ be their respective observed values where each $x_i$ takes on values from a fixed vocabulary $\mathbb{V} := \{1, \ldots, V\}$. Examples of these sequences include sentences where each letter or word is a single value, or streams of discrete events generated by some process or user. We will refer to individual variable-value pairs in the sequence as events.

We consider an autoregressive model $p_\theta(X_{1:N}) = \prod_{i=1}^{N} p_\theta(X_i | X_{1:i-1})$, parameterized by $\theta$ and trained on a given dataset of $M$ independent draws from a ground truth distribution $\mathbb{P}(X_{1:N})$. We assume that this model can be conditioned on a subsequence, termed the *history* $\mathcal{H}$. We will remap the indices of subsequent random variables to start at the first position[2]. We abbreviate conditioning on a history by an asterisk $*$, i.e., $\mathbb{P}^*(\cdot) := \mathbb{P}(\cdot | \mathcal{H})$ and $p_\theta^*(\cdot) := p_\theta(\cdot | \mathcal{H})$.

**Defining Probabilistic Queries** Given a specific history of events $\mathcal{H}$, there are a variety of different questions one could ask about the future continuing where the history left off: (Q1) What event is likely to happen next? (Q2) Which events are likely to occur $K > 1$ steps from now? (Q3) What is the distribution of when the next instance of $a \in \mathbb{V}$ occurs? (Q4) How likely is it that we will see event $a \in \mathbb{V}$ occur before $b \in \mathbb{V}$? (Q5) How likely is it for $a \in \mathbb{V}$ to occur $n$ times in the next $K$ steps?

---

[2]For example, if $|\mathcal{H}| = 3$ then $\mathbb{P}(X_4 | \mathcal{H})$ is the distribution of the 7th value in a sequence after conditioning on the first 3 values in the sequence.

| # | Question | Probabilistic Query | Cost $(K \cdot |\mathcal{Q}|)$ |
|---|----------|---------------------|-------------------------------|
| Q1 | Next event? | $p_\theta^*(X_1)$ | $\mathcal{O}(1)$ |
| Q2 | Event $K$ steps from now? | $p_\theta^*(X_K)$ | $\mathcal{O}(V^{K-1})$ |
| Q3 | Next instance of $a$? | $p_\theta^*(\tau(a) = K)$ | $\mathcal{O}((V-1)^{K-1})$ |
| Q4 | Will $a$ happen before $b$? | $p_\theta^*(\tau(a) < \tau(b))$ | $\mathcal{O}((V-2)^K)^\dagger$ |
| Q5 | How many instances of $a$ in $K$ steps? | $p_\theta^*(N_a(K) = n)$ | $\mathcal{O}\left(\binom{K}{n}(V-1)^{K-n}\right)$ |

Table 1: List of example questions, corresponding probabilistic queries, and associated costs of exact computation computation with an autoregressive model. The cost of accommodating a history $\mathcal{H}$ is assumed to be an additive constant for all queries. While Q4 extends to infinite time, the cost reported is for computing a lower bound up to $K$ steps.

We define a common framework for such queries by defining probabilistic queries to be of the form $p_\theta^*(X_{1:K} \in \mathcal{Q})$ with $\mathcal{Q} \subset \mathbb{V}^K$. This can be extended to the infinite setting (i.e., $p_\theta^*([X_k]_k \in \mathcal{Q})$ where $\mathcal{Q} \subset \mathbb{V}^\infty$). Exact computation of an arbitrary query is straightforward to represent:

$$p_\theta^*(X_{1:K} \in \mathcal{Q}) = \sum_{x_{1:K} \in \mathcal{Q}} p_\theta^*(X_{1:K} = x_{1:K}) = \sum_{x_{1:K} \in \mathcal{Q}} \prod_{k=1}^K p_\theta^*(X_k = x_k | X_{<k} = x_{<k}). \quad (1)$$

Depending on $|\mathcal{Q}|$, performing this calculation can quickly become intractable, motivating lower bounds or approximations (developed in more detail in Section 4). In this context it is helpful to impose structure on the query $\mathcal{Q}$ to make subsequent estimation easier, in particular by breaking $\mathcal{Q}$ into the following structured partition:

$$\mathcal{Q} = \cup_i \mathcal{Q}^{(i)} \text{ where } \mathcal{Q}^{(i)} \cap \mathcal{Q}^{(j)} = \emptyset \text{ for } i \neq j \quad (2)$$

$$\text{and } \mathcal{Q}^{(i)} = \mathcal{V}_1^{(i)} \times \mathcal{V}_2^{(i)} \times \cdots \times \mathcal{V}_K^{(i)} \text{ where } \mathcal{V}_k^{(i)} \subseteq \mathbb{V} \text{ for } k = 1, \ldots, K. \quad (3)$$

In words, this means a given query $\mathcal{Q}$ can be broken into a partition of simpler queries $\mathcal{Q}^{(i)}$ which take the form of a set cross product between restricted domains $\mathcal{V}_k^{(i)}$, one domain for each token $X_k$.[3] An illustration of an example query set can be seen in Fig. 2. A natural consequence of this is that:

$$p_\theta^*(X_{1:K} \in \mathcal{Q}) = \sum_i p_\theta^*\left(X_{1:K} \in \mathcal{Q}^{(i)}\right) = \sum_i p_\theta^*\left(\cap_{k=1}^K X_k \in \mathcal{V}_k^{(i)}\right),$$

which lends itself to more easily estimating each term in the sum. This will be discussed in Section 4.

**Queries of Interest** All of the queries posed earlier in this section can be represented under the framework detailed in Eq. (2) and Eq. (3), as illustrated in Table 1.

**Q1 & Q2** The queries $p_\theta^*(X_1 = a)$ and $p_\theta^*(X_K = a)$ for some $a \in \mathbb{V}$ can be represented with $\mathcal{Q} = \{a\}$ and $\mathcal{Q} = \mathbb{V}^{K-1} \times \{a\}$ respectively.

**Q3** The probability of the next instance of $a \in \mathbb{V}$ occurring at some point in time $K \geq 1$, $p_\theta^*(\tau(a) = K)$ where $\tau(\cdot)$ is the *hitting time*, can be represented as $\mathcal{Q} = (\mathbb{V} \setminus \{a\})^{K-1} \times \{a\}$. This can be adapted for a set $A \subset \mathbb{V}$ by replacing $\{a\}$ in $\mathcal{Q}$ with $A$.

**Q4** The probability of $a \in \mathbb{V}$ occurring before $b \in \mathbb{V}$, $p_\theta^*(\tau(a) < \tau(b))$, is represented as $\mathcal{Q} = \cup_{i=1}^\infty \mathcal{Q}^{(i)}$ where $\mathcal{Q}^{(i)} = (\mathbb{V} \setminus \{a, b\})^{i-1} \times \{a\}$. Lower bounds to this can be computed by limiting $i < i'$. Like Q3, this can also be extended to disjoint sets $A, B \subset \mathbb{V}$.

**Q5** The probability of $a \in \mathbb{V}$ occurring $n$ times in the next $K$ steps, $p_\theta^*(N_a(K) = n)$, is represented as $\mathcal{Q} = \cup_{i=1}^{C(K,n)} \mathcal{Q}^{(i)}$, where $N_a(K)$ is a random variable for the number of occurrences of events of type $a$ from steps 1 to $K$ and $\mathcal{Q}^{(i)}$'s are defined to cover all unique permutations of orders of products composed of: $\{a\}^n$ and $(\mathbb{V} \setminus \{a\})^{K-n}$. Like above, this can easily be extended for $A \subset \mathbb{V}$.

---

[3]Ideally, the partitioning is chosen to have the smallest number of $\mathcal{Q}^{(i)}$'s needed.

**Query Complexity**    From Eq. (1), exact computation of a query involves computing $K \cdot |Q|$ conditional distributions (e.g., $p_\theta^*(X_k|X_{<k} = x_{<k})$) in an autoregressive manner. Under the structured representation, the number of conditional distributions needed is equivalently $\sum_i \prod_{k=1}^{K} |\mathcal{V}_k^{(i)}|$. Non-attention based neural sequence models often define $p_\theta^*(X_k|X_{<k} = x_{<k}) := f_\theta(h_k)$ where $h_k = \text{RNN}_\theta(h_{k-1}, x_{k-1})$. As such, the computation complexity for any individual conditional distribution remains constant with respect to sequence length. We will refer to the complexity of this atomic action as being $\mathcal{O}(1)$. Naturally, the actual complexity depends on the model architecture and has a multiplicative scaling on the cost of computing a query. The number of atomic operations needed to exactly compute Q1-Q5 for this class of models can be found in Table 1. Should $p_\theta$ be an attention-based model (e.g., a transformer [Vaswani et al., 2017]) then the time complexity of computing a single one-step-ahead distribution becomes $\mathcal{O}(K)$, further exacerbating the **exponential growth** of many queries. Note that with some particular parametric forms of $p_\theta$, query computation can be more efficient, e.g., see Appendix C for a discussion on query complexity for Markov models.

## 4    Query Estimation Methods

Since exact query computation can scale exponentially in $K$ it is natural to consider approximation methods. In particular we focus on importance sampling, beam search, and a hybrid of both methods. All methods will be based on a novel proposal distribution, discussed below.

### 4.1    Proposal Distribution

For various estimation methods which will be discussed later, it is beneficial to have a proposal distribution $q(X_{1:K} = x_{1:K})$ whose domain matches that of the query $\mathcal{Q}$. For importance sampling, we will need this distribution as a proposal distribution, while we use it as our base model for selecting high-probability sequences in beam search. We would like the proposal distribution to resemble our original model while also respecting the query. One thought is to have $q(X_{1:K} = x_{1:K}) = p_\theta^*(X_{1:K} = x_{1:K}|X_{1:K} \in \mathcal{Q})$. However, computing this probability involves normalizing over $p_\theta^*(X_{1:K} \in \mathcal{Q})$ which is exactly what we are trying to estimate in the first place. Instead of restricting the *joint* distribution to the query, we can instead restrict every *conditional* distribution to the query's restricted domain. To see this, we first partition $\mathcal{Q} = \cup_i \mathcal{Q}^{(i)}$ and define an autoregressive proposal distribution for each $\mathcal{Q}^{(i)} = \prod_{k=1}^{K} \mathcal{V}_k^{(i)}$ as follows:

$$q^{(i)}(X_{1:K} = x_{1:K}) = \prod_{k=1}^{K} p_\theta^* \left( X_k = x_k | X_{<k} = x_{<k}, X_k \in \mathcal{V}_k^{(i)} \right) \tag{4}$$

$$= \prod_{k=1}^{K} \frac{p_\theta^*(X_k = x_k | X_{<k} = x_{<k}) \, \mathbb{1} \left( x_k \in \mathcal{V}_k^{(i)} \right)}{\sum_{v \in \mathcal{V}_k^{(i)}} p_\theta^*(X_k = v | X_{<k} = x_{<k})}$$

where $\mathbb{1}(\cdot)$ is the indicator function. That is, we constrain the outcomes of each conditional probability to the restricted domains $\mathcal{V}_k^{(i)}$ and renormalize them accordingly. To evaluate the proposal distribution's probability, we multiply all conditional probabilities according to the chain rule. Since the entire distribution is computed for a single model call $p_\theta^*(X_k|X_{<k} = x_{<k})$, it is possible to both sample a $K$-length sequence and compute its likelihood under $q^{(i)}$ with only $K$ model calls. Thus, we can efficiently sample sequences from a distribution that is both informed by the underlying model $p_\theta$ and that respects the given domain $\mathcal{Q}$. As discussed in the next section, this proposal will be used for importance sampling and for the base distribution on which beam search is conducted.

### 4.2    Estimation Techniques

**Sampling**    One can naively sample any arbitrary probability value using Monte Carlo samples to estimate $p_\theta^*(X_{1:K} \in \mathcal{Q}) = \mathbb{E}_{p_\theta^*}[\mathbb{1}(X_{1:K} \in \mathcal{Q})]$; however, this typically will have high variance. This can be substantially improved upon by exclusively drawing sequences from the query space $\mathcal{Q}$. Arbitrary queries can be written as a sum of probabilities of individual sequences, as seen in Eq. (1). This summation can be equivalently written as an expected value,

$$p_\theta^*(X_{1:K} \in \mathcal{Q}) = \sum_{x_{1:K} \in \mathcal{Q}} p_\theta^*(X_{1:K} = x_{1:K}) = |\mathcal{Q}| \, \mathbb{E}_{x_{1:K} \sim \mathcal{U}(\mathcal{Q})} \left[ p_\theta^*(X_{1:K} = x_{1:K}) \right],$$

where $\mathcal{U}$ is a uniform distribution. It is common for $p_\theta^*$ to concentrate most of the available probability mass on a small subset of the total possible space $\mathbb{V}^K$. Should $|\mathcal{Q}|$ be large, then $|\mathcal{Q}|\, p_\theta^*(X_{1:K} = x_{1:K})$ will have very large variance for $x_{1:K} \sim \mathcal{U}(\mathcal{Q})$. One way to mitigate this is to use importance sampling with our proposal distribution $q$ (Section 4.1):

$$p_\theta^*(X_{1:K} \in \mathcal{Q}) = |\mathcal{Q}|\, \mathbb{E}_{x_{1:K} \sim \mathcal{U}(\mathcal{Q})}\left[p_\theta^*(X_{1:K} = x_{1:K})\right] = \mathbb{E}_{x_{1:K} \sim q}\left[\frac{p_\theta^*(X_{1:K} = x_{1:K})}{q(X_{1:K} = x_{1:K})}\right]$$

$$\approx \frac{1}{M} \sum_{m=1}^{M} \frac{p_\theta^*\left(X_{1:K} = x_{1:K}^{(m)}\right)}{q\left(X_{1:K} = x_{1:K}^{(m)}\right)} \qquad \text{for } x_{1:K}^{(1)}, \ldots, x_{1:K}^{(M)} \stackrel{iid}{\sim} q.$$

It is worth noting that this estimator could be further improved by augmenting the sampling process to produce samples without replacement from $q$ (e.g., [Meister et al., 2021, Kool et al., 2019, Shi et al., 2020]); in this paper we restrict the focus to sampling with replacement.

**Search** An alternative to estimating a query by sampling is to instead produce a lower bound,

$$p_\theta^*(X_{1:K} \in \mathcal{Q}) = \sum_{x_{1:K} \in \mathcal{Q}} p_\theta^*(X_{1:K} = x_{1:K}) \geq \sum_{x_{1:K} \in \mathcal{B}} p_\theta^*(X_{1:K} = x_{1:K}),$$

where $\mathcal{B} \subset \mathcal{Q}$. In many situations, only a small subset of sequences $x_{1:K}$ in $\mathcal{Q}$ have a non-negligible probability of occurring due to the vastness of the total path space $V^K$. As such, it is possible for $|\mathcal{B}| \ll |\mathcal{Q}|$ while still having a minimal gap between the lower bound and the actual query value.

One way to produce a set $\mathcal{B} \subset \mathcal{Q}$ is through beam search. To ensure that beam search only explores the query space, instead of searching with $p_\theta$, we utilize $q$ for ranking beams. Since beam search is a greedy algorithm and for a given conditional $q(X_k = a|X_{<k}) \propto p_\theta^*(X_k = a|X_{<k})$ for $a \in \mathcal{V}_k$, the rankings will both respect the domain and be otherwise identical to using $p_\theta$ to rank. Typically, the goal of beam search is to find the most likely completion of a sequence without having to explore the entire space of possible sequences. This is accomplished by greedily selecting the top-$B$ most likely next step continuations, or *beams*, at each step into the future. Rather than finding a few high-likelihood beams, we are more interested in accumulating a significant amount of probability mass and less interested in the specific quantity of beams collected.

Traditional beam search has a fixed beam size $B$; however, this is not ideal for accumulating probability mass. As an alternative we develop *coverage-based* beam search where at each step in a sequence we restrict the set of beams being considered not to the top-$B$ but rather to the smallest set of beams that collectively exceed a predetermined probability mass $\alpha$, referred to as the "coverage".[4] More specifically, let $\mathcal{B}_k \subset \{x_{1:k} | x_{1:K} \in \mathcal{Q}\}$ be a set containing $|\mathcal{B}_k|$ beams for subsequences of length $k$. For brevity, we will assume that $\mathcal{Q} = \mathcal{V}_1 \times \cdots \times \mathcal{V}_K$.[5] $\mathcal{B}_{k+1}$ is a subset of $\mathcal{B}_k \times \mathcal{V}_{k+1}$ and is selected specifically to minimize $|\mathcal{B}_{k+1}|$ such that $q(X_{1:k+1} \in \mathcal{B}_{k+1}) \geq \alpha$. It can be shown that $p_\theta^*(X_{1:K} \in \mathcal{Q}) - p_\theta^*(X_{1:K} \in \mathcal{B}_K) \leq 1 - q(X_{1:K} \in \mathcal{B}_K)$ (and is often significantly less). See Appendix B for a proof.

There is one slight problem with having $\alpha$ be constant throughout the search. Since we are pruning based on the joint probability of the entire sequence, any further continuations of $\mathcal{B}_k$ will reduce the probability $q(X_{1:k+1} \in \mathbb{B}_{k+1})$. This may lead to situations in which every possible candidate sequence is kept in order to maintain minimal joint probability coverage. This can be avoided by filtering by $\alpha_k$ where $\alpha_1 > \cdots > \alpha_K = \alpha$, e.g., the geometric series $\alpha_k = \alpha^{k/K}$.

**A Hybrid Approach** Importance sampling produces an unbiased estimate, but can still experience large variance in spite of a good proposal distribution $q$ when $p_\theta$ is a heavy tailed distribution. Conversely, the beam search lower bound can be seen as a biased estimate with zero variance. We can remedy the limitations of both methods by recognizing that since $p_\theta^*(X_{1:K} \in \mathcal{Q}) = \sum_{x_{1:K} \in \mathcal{B}_K} p_\theta^*(X_{1:K} = x_{1:K}) + \sum_{x_{1:K} \in \mathcal{Q} \setminus \mathcal{B}_K} p_\theta^*(X_{1:K} = x_{1:K})$, where $\mathcal{B}_K$ is the set of sequences resulting from beam search, we can use importance sampling on the latter summation. The only caveat to this is that the proposal distribution must match the same domain of the summation: $\mathcal{Q} \setminus \mathcal{B}_K$.

---

[4]This is similar to the distinction between top-$K$ and top-$p$ / nucleus sampling commonly used for natural language generation [Holtzman et al., 2019].

[5]If $\mathcal{Q}$ requires partitioning into multiple $\mathcal{Q}^{(i)}$'s, we apply beam search to each sub query $p_\theta^*(X_{1:K} \in \mathcal{Q}^{(i)})$.

The proposal distribution we use to accomplish this is $q(X_{1:K} = x_{1:K} | X_{1:K} \notin \mathcal{B}_K)$ (see Eq. 4). This is implemented by storing all intermediate distributions (both original $p_\theta$ and proposal $q$) found during beam search and arranging them into a tree structure. All leaf nodes that are associated with $x_{1:K} \in \mathcal{B}_K$ have their transition probability zeroed out and then the effect of restriction and normalization is propagated up the tree to their ancestor nodes (details provided in Appendix D and E). Sampling from this new proposal distribution involves sampling from this normalized tree until we reach either an existing $K$-depth leaf node, in which case we are done, or a $< K$-depth leaf node in which case we complete the rest of the sequence by sampling from $q$ in Eq. (4) as usual.

Lastly, since our ultimate goal is to sample from the long tail of $p_\theta$, targeting a specific coverage $\alpha$ during beam search is no longer effective since achieving meaningfully large coverage bounds for non-trivial path spaces is generally intractable. Instead, we propose *tail-splitting* beam search to better match our goals. Let $w_k^{(i)} = p_\theta^*(X_{1:k+1} = x_{1:k+1}^{(i)})$ for $x_{1:k+1}^{(i)} \in \mathcal{B}_k \times \mathcal{V}_{k+1}$ such that $w_k^{(i)} \geq w_k^{(j)}$ if $i < j$. In this regime, $\mathcal{B}_{k+1} = \{x_{1:k+1}^{(i)}\}_{i=1}^B$ where $B = \arg\min_b \sigma(w_k^{(1:b)}) + \sigma(w_k^{(b+1:|\mathcal{B}_k \times \mathcal{V}_{k+1}|)})$. $\sigma(w_k^{(u:v)})$ is the empirical variance of $w_k^{(i)}$ for $i = u, \ldots, v$. This can be seen as performing 2-means clustering on the $w_k$'s and taking the cluster with the higher cumulative probability.

### 4.3 Saving Computation on Multiple Queries

Should multiple queries need to be performed, such as $p_\theta^*(\tau(a) = k)$ for multiple values of $k$, then there is potential to be more efficient in computing estimates for all of them. The feasibility of reusing intermediate computations is dependent on the set of queries considered. For simplicity, we will consider two base queries $\mathcal{Q} = \mathcal{V}_1 \times \cdots \times \mathcal{V}_K$ and $\mathcal{Q}' = \mathcal{V}_1' \times \cdots \times \mathcal{V}_{K'}'$ where $K < K'$. Due to the autoregressive nature of $p_\theta$, if $\mathcal{V}_i = \mathcal{V}_i'$ for $i = 1, \ldots, K-1$ then all of the distributions and sequences needed for estimating $p_\theta^*(X_{1:K} \in \mathcal{Q})$ are guaranteed to be intermediate results found when estimating $p_\theta^*(X_{1:K'} \in \mathcal{Q}')$. To be explicit, when estimating the latter query with beam search the intermediate $\mathcal{B}_K$ is the same as what would be directly computed for the former query. Likewise, for importance sampling if $x_{1:K'} \sim q(X_{1:K'})$ using Eq. (4) then the subsequence $x_{1:K} \sim q(X_{1:K})$. This does not apply when the sample path domain is further restricted, such as with the hybrid approach, in which case we cannot directly use intermediate results to compute other queries for "free."

## 5 Experiments and Results

**Experimental Setting** We investigate the quality of estimates of hitting time queries across various datasets, comparing beam search, importance sampling, and the hybrid method. We find that hybrid systematically outperforms both pure search and sampling given a comparable computation budget across queries and datasets. We also investigate the dependence of all three methods on the model entropy.

It is worth noting that we focus almost exclusively on hitting time queries in our primary experiments, as more complex queries *often decompose into operations over individual hitting times*. For example. consider the following decomposition of the "A before B" query (Q4):

$$p_\theta^*(\tau(a) < \tau(b)) = \sum_{k=1}^\infty p_\theta^*(\tau(a) = k, \tau(b) > k) = \sum_{k=1}^\infty p_\theta^*\big(X_k = a, X_{<k} \in (\mathbb{V} \setminus \{a, b\})^{k-1}\big)$$

Decompositions of other queries and their impact on approximation error are found in Appendix A.

**Datasets** We evaluate our query estimation methods on three user behavior and two language sequence datasets. **Reviews** contains sequences of Amazon customer reviews for products belonging to one of $V = 29$ categories [Ni et al., 2019]; **Mobile Apps** consists of app usage records over $V = 88$ unique applications [Aliannejadi et al., 2021]; **MOOCs** consists of student interaction with online course resources over $V = 98$ actions [Kumar et al., 2019]. We also use the works of William **Shakespeare** [Shakespeare] by modeling the occurrence of $V = 67$ unique ASCII characters. Lastly, we examine **WikiText** [Merity et al., 2017] to explore word-level sequence modeling applications with GPT-2, a large-scale language model with a vocabulary of $V = 50257$ word-pieces [Radford et al., 2019, Wu et al., 2016]. After preprocessing, none of the datasets contain personal identifiable information.

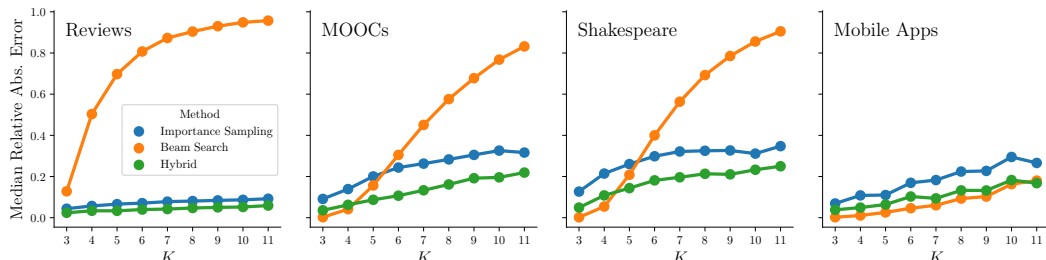

Figure 3: Median relative absolute error (RAE) between estimated probability and (surrogate) ground truth for $p_\theta^*(\tau(\cdot) = K)$ for importance sampling, beam search, and the hybrid method. As query path space grows with $K$, beam search quickly fails to bound ground truth while sampling remains robust, with the hybrid consistently outperforming all other methods, especially for large values of $K$. Ground truth values used to determine error are exact for $K \leq 4$ and approximated otherwise.

**Base Models**  While our proposed methods are amenable to autoregressive models of arbitrary structure, we focus our analysis specifically on recurrent neural networks. For all datasets except WikiText, we train Long-short Term Memory (LSTM) networks until convergence. To explore modern applications of sequence modeling with WikiText, we utilize GPT-2 with pre-trained weights from HuggingFace [Wolf et al., 2020]. Model training and experimentation utilized roughly 200 NVIDIA GeForce 2080ti GPU hours. Please refer to Appendix F for additional details.

**Experimental Methods**  We investigate computation-accuracy trade-offs between 3 estimation methods (beam search, importance sampling, and the hybrid) across all datasets. Query histories $\mathcal{H}$ are defined by randomly sampling $N = 1000$ sequences from the test split for all datasets except WikiText, from which we sample only $N = 100$ sequences due to computational limitations. For each query history and method, we compute the hitting time query estimate $p_\theta^*(\tau(a) = K)$ over $K = 3, \ldots, 11$, with $a$ determined by the $K^{th}$ symbol of the ground truth sequence.

To ensure an even comparison of query estimators, we fix the computation budget per query in terms of model calls $f_\theta(h_k)$ to be equal across all 3 methods, repeating experiments for different budget magnitudes roughly corresponding to $O(10), O(10^2), O(10^3)$ model calls (see Appendix H for full details). We intentionally select relatively small computation budgets per query to support systematic large-scale experiments over multiple queries up to relatively large values of $K$. Results for queries with GPT-2 are further restricted because of computational limits and are reported separately below.

To evaluate the accuracy of the estimates for each query and method, we compute the true probability of $K$ using exact computation for small values of $K \leq 4$. For larger values of $K$, we run importance sampling with a large number of samples $S$, where $S$ is adapted per query to ensure the resulting unbiased estimate has an empirical variance less than $\epsilon \ll 1$ (see Appendix F.3). This computationally-expensive estimate is then used as a surrogate for ground truth in error calculations.

Coverage-based beam search is not included in our results: we found that it experiences exponential growth with respect to $K$ and does not scale efficiently due to its probability coverage guarantees. Additional details are provided in Appendix H.3.

**Results: Accuracy and Query Horizon**  Using the methodology described above, for each query, we compute the relative absolute error (RAE) $|p - \hat{p}|/p$, where $\hat{p}$ is the estimated query probability generated by a particular method and $p$ is the ground truth probability or the surrogate estimate using importance sampling. For each dataset, for each of the 3 levels of computation budget, for each value of $K$, this yields $N = 1000$ errors for the $N$ queries for each method.

Fig. 3 shows the median RAE of the $N$ queries, per method, as a function of $K$, for each dataset, using the medium computation budget in terms of model calls. Across the 4 datasets the error increases systematically as $K$ increases. However, beam search is significantly less robust than the other methods for 3 of the 4 datasets: the error rate increases rapidly compared to importance sampling and hybrid. Beam search is also the most variable across datasets relative to other methods. The hybrid method systematically outperforms importance sampling along across all 4 datasets and for all values of $K$. In Appendix H we provide additional results; for the lowest and highest levels of computational

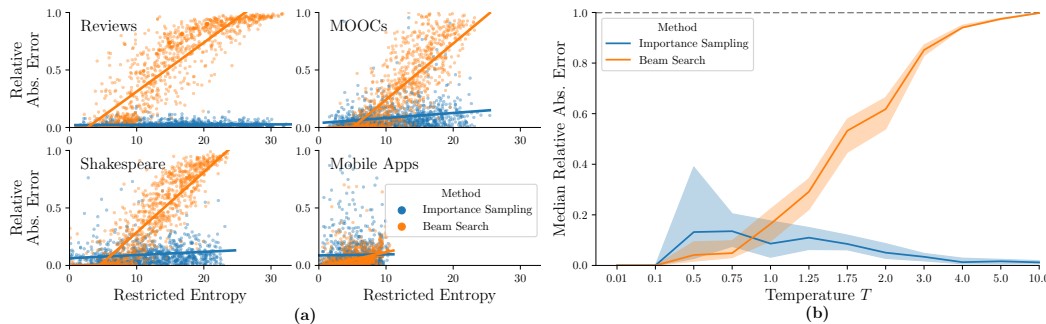

Figure 4: (left) RAE vs restricted entropy per query (with best linear fits), (right) Median RAE versus model temperature $T$ for Mobile App data. All errors computed using the same queries as in Fig. 3. Beam search errors correlate highly with model entropy even with the low-entropy Mobile Apps dataset, where increasing temperature $T$ directly induces this failure mode.

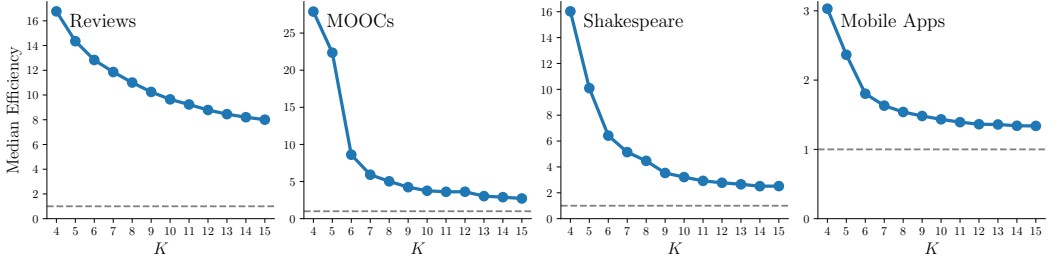

Figure 5: Median relative efficiency (over 1000 query histories and all vocabulary terms) of importance sampling estimation of the $K$-step marginal distributions for each dataset. The gray, dotted line represents 100% relative efficiency defined by naive query estimation. Relative efficiency is documented for $4 \leq K \leq 15$ to highlight the regime where ground truth cannot be tractably computed.

budget, for mean (instead of median) RAEs, and scatter plots for specific values of $K$ with more detailed error information. The qualitative conclusions are consistent across all experiments.

**Results: The Effect of Model Entropy**    We conjecture that the entropy of the proposal distribution $q$ conditioned on a given history $H_q^*(X_{1:K}) = -\mathbb{E}_{x_{1:K} \sim q}[\log q(X_{1:K} = x_{1:K})]$, which we refer to as *restricted entropy*, is a major factor in the performance of the estimation methods. Fig. 4(a) shows the RAE per query (with a linear fit) as a function of estimated restricted entropy for importance sampling and beam search. The results clearly show that entropy is driving the query error in general and that the performance of beam search is much more sensitive to entropy than sampling. The difference in entropy characteristics across datasets explains the differences in errors we see in Fig. 3. In particular, the Mobile Apps dataset is in a much lower entropy regime than the other three datasets.

To further investigate the effect of entropy, we alter each model by applying a temperature $T > 0$ to every conditional factor: $p_{\theta,T}(X_k|X_{<k}) \propto p_\theta(X_k|X_{<k})^{1/T}$, effectively changing the entropy ranges for the models. Fig. 4(b) shows the median RAE, for query $p_{\theta,T}^*(\tau(\cdot) = 4)$, as a function of model temperature for the Mobile Apps data. As predicted from Fig. 4(a), the increase in $T$, and corresponding increase in entropy, causes beam search's error to converge to 1, while the sampling error goes to 0. As $T$ increases, each individual sequence will approach having $1/|\mathcal{Q}|$ mass, thus needing many more beams to have adequate coverage. Results for other queries and the other three datasets (in Appendix H) further confirm the fundamental bifurcation of error between search and sampling (that we see in Fig. 4(b)) as a function of entropy.

**Results: Relative Efficiency of Proposal Distribution over Naive Query Estimation**    We also examine the relative efficiency improvements of our proposal distribution against naive Monte Carlo sampling:

$$p_\theta^*(X_{1:K} \in \mathcal{Q}) = \mathbb{E}_{x_{1:K} \sim p^*}[\mathbb{1}(x_{1:K} \in \mathcal{Q})]$$

Our relative efficiency calculations in Fig. 5 represent the variance ratio of naive query estimates and estimates from our query proposal distribution. As shown, all datasets witness improvement over naive sampling efficiency and often by a significant margin. We also observe that relative efficiency is largest for shorter query horizons, approaching naive sampling efficiency as $K$ increases.

**Results: Queries with a Large Language Model**  We further explore entropy's effect on query estimation with GPT-2 and the WikiText dataset for $N = 100$, $K = 3, 4$, across 3 computation budgets. With a vocabulary 500x larger than the other models, GPT-2 allows us to examine queries relevant to NLP applications. The resulting high entropy causes beam search to fail to match surrogate ground truth given the computation budgets (consistent with earlier experiments), with a median RAE of 82% (for $K = 4$ and a budget of $O(10^3)$). By contrast, importance sampling's median RAE under the same setting is 13%, **a 6x reduction**. Additional results are in Appendix H.5.

## 6  Discussion

**Future Directions:**  This work provides a starting point for multiple different research directions beyond the scope of this paper. One such direction is exploring more powerful search and sampling methods for query estimation. This includes methods for sampling without replacement, such as the Gumbel-max method [Huijben et al., 2022, to appear], sampling importance-resampling (SIR) [Skare et al., 2003], as well as new heuristics for automatically trading off search and sampling on a per-query basis. Another direction for exploration is amortized methods, such as learning models before queries are issued, that are specifically designed to help answer queries. Learning models that include queries as part of regularization terms in objective functions can also build on this work, e.g., learning models that don't only rely on one-step-ahead losses [Meister and Cotterell, 2021]. The work in this paper has also recently been broadened to continuous-time models for marked temporal point processes (requiring marginalization over time as well as event types) [Boyd et al., 2022].

**Limitations:**  Our results rely on only four datasets with a single autoregressive neural sequence model trained for each, naturally limiting the breadth of conclusions that we can draw. However, given that the results are consistent and validated by algorithm-independent entropy and efficiency analyses, we believe these findings have general validity and provide a useful starting point for others interested in the problem of querying neural sequence models. Another potential limitation is that our four datasets have relatively small vocabularies ($V = O(10^2)$, small by NLP standards at least); this choice was largely driven by computational limitations in terms of being able to conduct conclusive experiments (e.g. averaging over many event histories). Our (limited) results with GPT-2 provide clues on what may be achievable with much larger vocabularies: systematic analysis of querying for such models is a natural target for future work. Our work also does not address the issue of model inaccuracy: our results are entirely focused on computing query estimates in terms of a model's distribution instead of the data-generating process. Exploring the effect of miscalibration errors in autoregressive models on $k$-step-ahead query estimates is a promising avenue for future work.

**Potential Negative Societal Impacts:**  Since the focus of this work is making predictions with data that is typically generated by individuals (language, online activity), there is naturally a potential for abuse of such methods. For example, if the underlying model in a system is miscalibrated, decisions could be made that negatively impact individuals, e.g., recommending a student be dropped from an online course if the model incorrectly predicts they will not participate in future course modules. Even if the underlying model is well-calibrated, predictive queries could potentially be used in a proactive manner to bias decisions against individuals whose event sequences are atypical, e.g., in a chatbot context, inaccurately predicting future language patterns for certain individuals, leading to interruption and generation of an inappropriate response.

**Acknowledgements:**  We thank the NeurIPS reviewers for their suggestions on improving the original version of this paper. This work was supported by National Science Foundation Graduate Research Fellowship grant DGE-1839285 (AB and SS), by an NSF CAREER Award (SM), by the National Science Foundation under award numbers 1900644 (PS), 2003237 (SM), and 2007719 (SM), by the National Institute of Health under awards R01-AG065330-02S1 and R01-LM013344 (PS), by the Department of Energy under grant DE-SC0022331 (SM), by the HPI Research Center in Machine Learning and Data Science at UC Irvine (SS), and by Qualcomm Faculty awards (PS and SM).

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
