# Appendix to: Predictive Querying for Autoregressive Neural Sequence Models

**Alex Boyd**[* 1]    **Sam Showalter**[* 2]    **Stephan Mandt**[1,2]    **Padhraic Smyth**[1,2]
[1]Department of Statistics    [2]Department of Computer Science
University of California, Irvine
{alexjb,showalte,mandt,p.smyth}@uci.edu

## A    Query Decomposition (Section 3 in paper)

All of the questions posed in Table 1 in the main paper can be decomposed into readily available components that our model $p_\theta$ can estimate. Should we not utilize the proposed query structure $(\mathcal{Q} = \cup_i \mathcal{V}_1^{(i)} \times \cdots \times \mathcal{V}_K^{(i)})$ then these are the derivations that would need to be performed individually in order to compute these values exactly on a case-by-case basis.

**Q1**    $\mathbb{P}^*(X_1)$ is already naturally in a form that our model can directly estimate due to the autoregressive factorization imposed by the architecture: $p_\theta^*(X_1)$. Furthermore, if we are interested in any potential continuing sequence $X_{1:K} = x_{1:K}$ this can easily be computed via $p_\theta^*(X_{1:K} = x_{1:K}) = \prod_{k=1}^{K} p_\theta^*(X_k = x_k | X_{<k} = x_{<k})$.

**Q2**    To evaluate $p_\theta^*(X_K)$, terms $X_{<K}$ need to be marginalized out. This is naturally represented like so:

$$p_\theta^*(X_K) = \mathbb{E}_{x_{<K} \sim p_\theta^*(X_{<K})} \left[ p_\theta^*(X_K | X_{<K} = x_{<K}) \right]$$
$$= \sum_{x_{<K} \in \mathbb{V}^{K-1}} p_\theta^*(X_K, X_{<K} = x_{<K})$$

It is helpful to show both the exact summation form as well as the expected value representation as both will be useful in Section 4.

**Q3**    The "hitting time" or the next occurrence of a specific event type $a \in \mathbb{V}$ is defined as $\tau(a)$. Evaluating the distribution of the hitting time with our model can be done like so:

$$p_\theta^*(\tau(a) = K) = p_\theta^*(X_K = a, X_{<K} \neq a),$$
$$= \sum_{x_{<K} \in \{\mathbb{V} \setminus \{a\}\}^{K-1}} p_\theta^*(X_K = a, X_{<K} = x_{<K}).$$

The value $a \in \mathbb{V}$ can be easily replaced with a set of values $A \subset \mathbb{V}$ in these representations.

$$p_\theta^*(\tau(A) = k) = p_\theta^*(X_K \in A, X_{<K} \in (\mathbb{V} \setminus A)^{K-1}),$$
$$= \sum_{x_{<K} \in \{\mathbb{V} \setminus A\}^{K-1}} \sum_{a \in A} p_\theta^*(X_K = a, X_{<K} = x_{<K})$$

Interestingly, we can see that Q3 is a generalization of Q2 by noting that they are identical when $A = \{\}$.

---

[*]Authors contributed equally

36th Conference on Neural Information Processing Systems (NeurIPS 2022).

**Q4**   To evaluate $p_\theta^*(\tau(a) < \tau(b))$, we must consider the possible instances where this condition is fulfilled. In doing so, we end up evaluating multiple queries similar in form to Q3, like so:

$$p_\theta^*(\tau(a) < \tau(b)) = \sum_{k=1}^{\infty} p_\theta^*(\tau(a) = k, \tau(b) > k)$$

$$= \sum_{k=1}^{\infty} p_\theta^*(X_k = a, X_{<k} \in (\mathbb{V} \setminus \{a, b\})^{k-1}) \tag{11}$$

In practice, computing this exactly is intractable due to it being an infinite sum. There are two potential approaches one could take to subvert this. The first of which is to ask a slightly different query:

$$p_\theta^*(\tau(a) < \tau(b) | \tau(a) \le K) = \frac{p_\theta^*(\tau(a) < \tau(b), \tau(a) \le K)}{p_\theta^*(\tau(a) \le K, \mathcal{H}_T)}$$

$$= \frac{\sum_{k=1}^{K} p_\theta^*(X_k = a, X_{<k} \in (\mathbb{V} \setminus \{a, b\})^{k-1})}{\sum_{k=1}^{K} p_\theta^*(\tau(a) = k)}.$$

The other option is to produce a lower bound on this expression by evaluating the sum in Eq. (11) for the first $K$ terms. We can achieve error bounds on this estimate by noting that $p_\theta^*(\tau(a) < \tau(b)) + p_\theta^*(\tau(a) > \tau(b)) = 1$. As such, if we evaluate Eq. (11) up to $K$ terms for both $p_\theta^*(\tau(a) < \tau(b))$ and $p_\theta^*(\tau(b) < \tau(a))$, the difference between the sums will be the maximum error either lower bound can have.

Similar to Q3, we can also ask this query with sets $A \perp B \subset \mathbb{V}$ instead of values $a, b$.

$$p_\theta^*(\tau(A) < \tau(B)) = \sum_{k=1}^{\infty} p_\theta^*(\tau(A) = k, \tau(B) > k)$$

$$= \sum_{k=1}^{\infty} \sum_{a \in A} p_\theta^*(X_k = a, X_{<k} \in (\mathbb{V} \setminus (A \cup B))^{k-1})$$

**Q5**   Evaluating $p_\theta^*(N_a(K) = n)$ also involves decomposing this value of interest into statements involving hitting times.

Let $\tau^{(l)}(a)$ be the $l^{\text{th}}$ hitting time of a specific event of interest $a$. In other words, the time of the $l^{\text{th}}$ occurrence of $a$. Assuming $n \le K$:

$$p_\theta^*(N_a(K) = n)$$

$$= \sum_{i_1 < i_2 < \cdots < i_n \le K} p_\theta^*(\tau^{(1)}(a) = i_1, \tau^{(2)}(a) = i_2, \ldots, \tau^{(n)}(a) = i_n, \tau^{(n+1)}(a) > K)$$

$$= \sum_{i_1 < i_2 < \cdots < i_n \le K} p_\theta^*(X_{i_1} = \tau^{(1)}(a) = i_1, \tau^{(2)}(a) = i_2, \ldots, \tau^{(n)}(a) = i_n, \tau^{(n+1)}(a) > K)$$

**Approximation Errors**   With queries that fall under the Q5 category (e.g., how likely will the event $a$ happen $n$ times in the next $K$ steps? $p_\theta^*(N_a(K) = n)$), while in the limiting case C(K,n) will become the dominant scaling factor for the size of the query space, in many circumstances this actually isn't as impactful as one might expect. For instance, it can be shown that the query space of a Q5 query is larger than that of a hitting time query (Q3) evaluated one step further (at $K + 1$) if $\sqrt[n]{C(K, n)} > V$ where $V$ is the vocabulary size. Looking at our main set of experiments from Section 5, we can see that when evaluating hitting time queries up to 11 steps into the future ($K = 11$), these queries had a slightly larger query space of than that of $p_\theta^*(N_a(K) = n)$ for $K = 10, \forall_{0 \le n \le K}$ and thus can be seen as comparable in complexity. This behavior easily extends to larger values of $K$ (e.g., $K \approx 100$), especially for larger vocabulary sizes. This is due to the $n^{\text{th}}$ root dramatically slowing down the factorial growth in the inequality shared above.

# B  Proof for Coverage-Based Beam Search Error Upper Bound (Section 4.2 in paper)

**Proposition 1.** *For a given set of decoded sequences $\mathcal{B} \subset \mathcal{Q}$ with coverage $q(X_{1:K} \in \mathcal{B})$, the error between the true probabilistic query value $p_\theta^*(X_{1:K} \in \mathcal{Q})$ and the lower bound $p_\theta^*(X_{1:K} \in \mathcal{B})$ is bounded above by the complement of the coverage: $1 - q(X_{1:K} \in \mathcal{B})$.*

*Proof.* For brevity, we will assume that $\mathcal{Q} = \mathcal{V}_1 \times \cdots \times \mathcal{V}_K$ (although this can easily be extended to the general case). Since $q(X_k = x_k | X_{<k} = x_{<k}) = \frac{p_\theta^*(X_k = x_k | X_{<k} = x_{<k}) \mathbb{1}(x_k \in \mathcal{V}_k)}{p_\theta^*(X_k \in \mathcal{V}_k | X_{<k} = x_{<k})}$ and $p_\theta^*(\cdot) \leq 1$, it follows that $q(X_k = x_k | X_{<k} = x_{<k}) \geq p_\theta^*(X_k = x_k | X_{<k} = x_{<k})$. This becomes a strict inequality should $p_\theta^*(X_k \in \mathcal{V}_k | X_{<k} = x_{<k}) < 1$ (which is often the case should $\mathcal{V}_k \subset \mathbb{V}$). Since this holds for arbitrary $k$ and $x_k$, it then follows that $q(X_{1:K} = x_{1:K}) \geq p_\theta^*(X_{1:K} = x_{1:K})$ for any $x_{1:K} \in \mathcal{Q}$.[2] This inequality becomes strict should any sequence not in the query set have non-zero probability, i.e., $p_\theta^*(X_{1:K} = x_{1:K}) > 0$ for any $x_{1:K} \in \mathbb{V}^K \setminus \mathcal{Q}$.

The target value that we are estimating can be broken up into the following terms:

$$p_\theta^*(X_{1:K} \in \mathcal{Q}) = p_\theta^*(X_{1:K} \in \mathcal{B}) + p_\theta^*(X_{1:K} \in \mathcal{Q} \setminus \mathcal{B}).$$

Rearranging these terms yields us the error of our lower bound. We can easily derive an upper bound on the error for an arbitrary set $\mathcal{B}$:

$$
\begin{aligned}
p_\theta^*(X_{1:K} \in \mathcal{Q}) - p_\theta^*(X_{1:K} \in \mathcal{B}) &= p_\theta^*(X_{1:K} \in \mathcal{Q} \setminus \mathcal{B}) \\
&= \sum_{x_{1:K} \in \mathcal{Q} \setminus \mathcal{B}} p_\theta^*(X_{1:K} = x_{1:K}) \\
&\leq \sum_{x_{1:K} \in \mathcal{Q} \setminus \mathcal{B}} q(X_{1:K} = x_{1:K}) \\
&= q(X_{1:K} \in \mathcal{Q} \setminus \mathcal{B}) \\
&= q(X_{1:K} \in \mathcal{Q}) - q(X_{1:K} \in \mathcal{B}) \\
&= 1 - q(X_{1:K} \in \mathcal{B}) \leq 1 - \alpha
\end{aligned}
$$

where $\alpha$ is the targeted coverage probability used to find $\mathcal{B}$ with coverage-based beam search.[3] This inequality becomes strict should any sequence not in the query set have non-zero probability, i.e., $p_\theta^*(X_{1:K} = x_{1:K}) > 0$ for any $x_{1:K} \in \mathbb{V}^K \setminus \mathcal{Q}$. $\qquad\square$

# C  Complexity of Predictive Querying with First-Order Markov Models (Section 3 in paper)

As a point of reference for neural sequence models we summarize below the cost of various queries for Markov models (using basic results from the theory of finite Markov chains, e.g., see [Kemeny and Snell, 1983]). Consider a first-order ergodic homogeneous Markov chain with a $V \times V$ transition matrix with elements $p(x_{k+1} | x_k)$. We analyze below the complexity of various queries for such a chain, conditioned on the current observation $X_0 = v$, where $v \in \mathbb{V}$ (this is analogous to conditioning on $\mathcal{H}$ for neural sequence models). We use $p(\cdot | v)$ as shorthand for $p(\cdot | X_0 = v)$.

**Q2: computing $p(X_k | v)$**    This can be computed recursively by computing the conditional distribution $p(X_1 | v)$, then $p(X_2 | v) = \sum_{v'} p(X_2 | v') p(v' | v)$, and so on, resulting in $k$ matrix multiplications, with complexity $O(kV^2)$.

**Q3: computing $p(\tau(a) = k | v)$**    For finite $k$, we can compute the result for all values $k' \leq k$ using $k - 1$ matrix multiplications where, at each step $k' = 2, \ldots, k - 1$, all states (events) except $a$ are marginalized over. Thus, the complexity is $O(kV^2)$.

---

[2]This can be seen by comparing their autoregressive factorizations term by term.

[3]Note that $q(X_{1:K} \in \mathcal{Q}) = 1$ because by design $q(X_{1:K} = x_{1:K}) = 0$ for every $x_{1:K} \notin \mathcal{Q}$.

In general it is straightforward to show that

$$p(\tau(a) = k|v) = p(a|v), \quad k = 1$$

and

$$p(\tau(a) = k|v) = p(a|\bar{a})p(\bar{a}|v)p(\bar{a}|\bar{a})^{(k-2)}, \quad k > 1$$

where $p(a|\bar{a})$ is the probability of transitioning to state $a$ given that the chain is not in state $a$. Computation of $p(a|\bar{a})$ requires knowledge of the steady-state probabilities of the chain $\pi_a$ and $\pi_{\bar{a}} = \sum_{v' \neq a} \pi_{v'}$. Computing the steady-state probabilities requires inversion of a $V \times V$ matrix, with a complexity of $O(V^3)$. This general solution will be faster to compute than the version using matrix multiplication (up to horizon $k$) whenever $k > V$.

**Q4: computing** $p(\tau(a) < \tau(b)|v)$   Let $c$ be the set of all states (events) in $\mathbb{V}$ except for $a$ and $b$. It is straightforward to show that

$$p(\tau(a) < \tau(b)|v) = p(a|v) + p(c|v)\frac{p(a|c)}{p(a|c) + p(b|c)}$$

and

$$p(\tau(b) < \tau(a)|v) = p(b|v) + p(c|v)\frac{p(b|c)}{p(a|c) + p(b|c)}$$

where $p(a|c)$ and $p(b|c)$ are the probabilities of transitioning to $a$ and $b$, respectively, given that the system is currently in a state that is neither $a$ or $b$. Computing these probabilities again requires knowledge of the steady-state probabilities $\pi_a, \pi_b, \pi_c$, resulting in $O(V^3)$ time complexity.

**General Queries and Higher Order Markov Models**   For simplicity, we will assume that a query takes the form $\mathcal{Q} = \mathcal{V}_1 \times \cdots \times \mathcal{V}_K$ and we are interested in $p(X_{1:K} \in \mathcal{Q}|\mathcal{H})$ for a $m^{\text{th}}$-order Markov model $p$. This model can be defined with an $(m+1)$-dimensional tensor $\Pi \in \mathbb{R}^{V \times \cdots \times V}$ with elements $\pi_{i_1,\ldots,i_m,i_{m+1}}$ such that $\sum_{j=1}^{V} \pi_{i_1,\ldots,i_m,j} = 1$ for all $i_1, \ldots, i_m \in \mathbb{V}$. Alternatively, $\pi_{i_1,\ldots,i_m,i_{m+1}} = p(X_{j+m+1} = i_{m+1}|X_{j+1:j+m} = i_{1:m})$ for $j \geq 0$.

To marginalize out $X_{m+1}$ and compute the conditional distribution $p(X_{m+2}|X_{1:m})$, that requires the following computation:

$$p(X_{m+2} = x_{m+2}|X_{1:m} = x_{1:m}) = \sum_{v \in \mathbb{V}} p(X_{m+2} = a, X_{m+1} = v|X_{1:m} = x_{1:m})$$

$$= \sum_{v \in \mathbb{V}} p(X_{m+2} = x_{m+2}|X_{m+1} = v, X_{2:m} = x_{2:m})p(X_{m+1} = v|X_{1:m} = x_{1:m})$$

$$= \sum_{v \in \mathbb{V}} \pi_{x_2,\ldots,x_m,v,a}\pi_{x_1,\ldots,x_{m-1},x_m,v}$$

If we perform this over all values of $x_{1:m}$, we can construct a new transition tensor representing $p(X_{m+2}|X_{1:m})$. We will denote this new tensor as being equal to $\Pi \otimes \Pi$ where $(\Pi \otimes \Pi)_{i_1,\ldots,i_m,v} = p(X_{j+m+2} = v|X_{j+1:j+m} = i_{1:m})$. This operation has a computation complexity of $\mathcal{O}(V^{m+1})$.

This special product can be done repeatedly to further marginalize out. For instance, performing this operation on $\Pi$ $(n-1)$-times results in $(\Pi \otimes \cdots \otimes \Pi)_{i_1,\ldots,i_m,v} = p(X_{j+m+n} = v|X_{j+1:j+m} = i_{1:m})$, thus marginalizing out $n-1$ terms: $X_{j+m+1:j+m+n-1}$.

Transitioning into a restricted space $\mathcal{V}$ can be done easily by defining a restricted transition tensor $\Pi_{\mathcal{V}}$ such that $(\Pi_{\mathcal{V}})_{i_1,\ldots,i_m,v} \propto \pi_{i_1,\ldots,i_m,v}$ if $v \in \mathbb{V}$, otherwise $(\Pi_{\mathcal{V}})_{i_1,\ldots,i_m,v} = 0$ for all $i_1, \ldots, i_m, v \in \mathbb{V}$.

With this, we have everything we need to compute $p(X_{1:K} \in \mathcal{Q}|\mathcal{H})$. If the last $m$-values of the history $\mathcal{H}$ are equal to $i_1, \ldots, i_m$, then:

$$p(X_{1:K} \in \mathcal{Q}|\mathcal{H}) = \prod_{k=1}^{K} p(X_k \in \mathcal{V}_k | X_{<k} \in \mathcal{V}_1 \times \cdots \times \mathcal{V}_{k-1})$$

$$= \prod_{k=1}^{K} \sum_{v \in \mathcal{V}_k} p(X_k = v | X_{<k} \in \mathcal{V}_1 \times \cdots \times \mathcal{V}_{k-1})$$

$$= \prod_{k=1}^{K} \sum_{v \in \mathcal{V}_k} (\Pi_{\mathcal{V}_1} \otimes \cdots \otimes \Pi_{\mathcal{V}_k})_{i_1, \ldots, i_m, v}.$$

The dominant factor in the computational complexity of this is computing all of the special products of $\Pi$. As such, this has a total computational complexity of $\mathcal{O}((k-1)V^{m+1})$.

## D   Details for the Hybrid Method (Section 4 in paper)

The hybrid method estimates an arbitrary probabilistic query $p_\theta^*(X_{1:K} \in \mathcal{Q})$ by first using some variant of beam search (in our results we use our proposed *tail-splitting* beam search, see Section 3) to find a lower bound $p_\theta^*(X_{1:K} \in \mathcal{B})$, and then using a sampling method (we use importance sampling, see Section 3) to estimate the remainder $p_\theta^*(X_{1:K} \in \mathcal{Q} \setminus \mathcal{B})$. The only complicated portion of this process lies in how we derive an acceptable proposal distribution for the sampling phase that respects the domain $\mathcal{Q} \setminus \mathcal{B}$. The following paragraphs detail how we construct such a distribution.

**Viewing $\mathbb{V}^K$ and $p_\theta^*$ as Trees**   In the space of all possible sequences of length $K$, $X_{1:K} \in \mathbb{V}^K$, one can represent these sequences as paths in a tree. Each node in this tree represents a single element in a sequence $X_k \in \mathbb{V}$ with depth $k$, with parent and children nodes representing previous and potential future values in the sequence respectively. The root node either represents the very beginning of a sequence, or a concrete history $\mathcal{H}$ to condition on.

This tree can be augmented into a probabilistic one by defining edges between nodes as the conditional probability of a child node being next in a sequence, conditioned on all ancestors of that child. These probabilities are naturally determined by $p_\theta^*(X_k = x_k | X_{<k} = x_{<k})$ where $x_k$ is the child node value and $x_{<k}$ are the ancestors' values.

**Building the Tree**   Any subset of $\mathbb{V}^K$ can also be represented as a tree, and in fact will be a sub-tree of the one that represents $\mathbb{V}^K$. As such, there exists a tree that represents $\mathcal{Q}$. Our usual proposal distribution $q$ is a natural source of conditional probabilities for the edges. While none of these trees with their edge weights are fully known ahead of time, we do explore and uncover them through the process of beam search. As such, during the beam search phase of the hybrid method we keep track of any conditional distributions $q(X_k = x_k | X_{<k} = x_{<k})$ that are computed and use them to construct a partial view of the tree for $\mathcal{Q}$. Note that the end result of this process is a tree that will likely have many paths that do not fully reach depth $K$; however, there will be at least $|\mathcal{B}|$ many that do.

For our purposes, it is also useful to keep track of $p_\theta^*(X_k = x_k | X< k = x_{<k})$ over this restricted set, as well as model byproducts such as hidden states to reduce computation redundancy later.

**Pruning the Tree**   After beam search has completed, we are left with a resulting set of beams $\mathcal{B}$ and a probabilistic tree representing $q$. We would now like to alter this tree such that its weights represent $q_\mathcal{B}(X_{1:K} = x_{1:K}) := q(X_{1:K} = x_{1:K} | X_{1:K} \notin \mathcal{B})$. This alteration can be accomplished by adjusting the edge weights in the tree recursively as detailed below. New weight assignments will be denoted by $q'$ to differentiate from old weights $q$. The steps to the procedure are defined as follows:

1. At the final depth $K$, assign edge weights $q'(X_K = x_K | X_{<K} = x_{<K}) = 0$ for all $x_{1:K} \in \mathcal{B}$. All other edge weights in the final depth will have new weights $q'(X_K = x_K | X_{<K} = x_{<K}) = q(X_K = x_K | X_{<K} = x_{<K})$.
2. At the next layer above with depth $k = K - 1$, assign edge weights as $q'(X_k = x_k | X_{<k} = x_{<k}) = q(X_k = x_k | X_{<k} = x_{<k}) \sum_{v \in \mathbb{V}} q'(X_{k+1} = v | X_{\leq k} = x_{\leq k})$ for all sub-sequences $x_{1:k} \in \mathcal{B}$.

3. Repeat step 2 iteratively for $k = K - 2, K - 3, \ldots, 2, 1$.
4. Finally, normalize every conditional distribution for every node whose children edges were altered such that they each sum to 1.

After these steps are completed, $q_{\mathcal{B}}(X_{1:K} = x_{1:K}) = \prod_{k=1}^{K} q'(X_k = x_k | X_{<k} = x_{<k})$. Note that weights related to sequences that were not discovered during beam search, and are thus not in the tree, are not altered and still match the original proposal distribution $q$. As such, to sample sequences from the tree, we start at the root node and sample from each successive conditional distribution until either depth $K$ or a leaf node at depth $k < K$ is reached. In the former scenario, the sampling is complete. In the latter, the remaining values of the sequence are sampled from $q(\cdot | X_{\leq k} = x_{\leq k})$ like usual.

# E   Variance of Estimates from the Hybrid Method (Section 4 in paper)

We are assuming to be under the hybrid method regime where a collection of sequences $\mathcal{B} \subset \mathcal{Q}$ relevant to answering $p_\theta^*(X_{1:K} \in \mathcal{Q})$ have been deterministically found and are interested in using sampling methods to estimate the remainder $p_\theta^*(X_{1:K} \in \mathcal{Q} \setminus \mathcal{B})$. For brevity, we will assume that $\mathcal{Q} = \mathcal{V}_1 \times \cdots \times \mathcal{V}_K$. As mentioned in the previous section, we leverage our originally presented proposal distribution $q$ by further restricting the domain to $\mathcal{Q} \setminus \mathcal{B}$ in order to be used in this scenario. This will be represented by

$$
\begin{aligned}
q_{\mathcal{B}}(X_{1:K} = x_{1:K}) &:= q(X_{1:K} = x_{1:K} | X_{1:K} \notin \mathcal{B}) \\
&= \frac{q(X_{1:K} = x_{1:K}) \mathbb{1}(x_{1:K} \in \mathcal{Q} \setminus \mathcal{B})}{1 - q(X_{1:K} \in \mathcal{B})}.
\end{aligned}
$$

Note that an associated autoregressive form $q_{\mathcal{B}}(X_k = x_k | X_{<k} = x_{<k})$ exists and is well defined; however, the exact definition is a bit unwieldy. Please refer to the previous section for more details.

Using this proposal distribution, we can easily estimate the remaining probability:

$$
\begin{aligned}
p_\theta^*(X_{1:K} \in \mathcal{Q} \setminus \mathcal{B}) &= \mathbb{E}_{x_{1:K} \sim q_{\mathcal{B}}} \left[ \frac{p_\theta^*(X_{1:K} = x_{1:K})}{q_{\mathcal{B}}(X_{1:K} = x_{1:K})} \right] \\
&\approx \frac{1}{M} \sum_{m=1}^{M} \frac{p_\theta^*(X_{1:K} = x_{1:K}^{(m)})}{q_{\mathcal{B}}(X_{1:K} = x_{1:K}^{(m)})} \qquad \text{for } x_{1:K}^{(1)}, \ldots, x_{1:K}^{(M)} \overset{iid}{\sim} q_{\mathcal{B}}.
\end{aligned}
$$

Let $\omega_{\mathcal{B}}(x_{1:K}) := \frac{p_\theta^*(X_{1:K} = x_{1:K})}{q_{\mathcal{B}}(X_{1:K} = x_{1:K})}$ and for brevity we will refer to $p_\theta^*(X_{1:K} \in \mathcal{Q} \setminus \mathcal{B})$ as $\bar{\omega}_{\mathcal{B}}$.

If we assume $\mathcal{B}' = \mathcal{B} \cup \{\hat{x}_{1:K}\}$ for some $\hat{x}_{1:K} \in \mathcal{Q} \setminus \mathcal{B}$, then it is interesting to determine when exactly there will be a reduction in sampling variance for $p_\theta^*(X_{1:K} \in \mathcal{Q} \setminus \mathcal{B}')$ versus $p_\theta^*(X_{1:K} \in \mathcal{Q} \setminus \mathcal{B})$ as this will give insight into when the hybrid method is successful. In other words, we would like to show when the following inequality holds true:

$$
\Delta_{\text{Var}} := \text{Var}_{x_{1:K} \sim q_{\mathcal{B}}} \left[ \omega_{\mathcal{B}}(x_{1:K}) \right] - \text{Var}_{x_{1:K} \sim q_{\mathcal{B}'}} \left[ \omega_{\mathcal{B}'}(x_{1:K}) \right] \geq 0
$$

If this is true for a given $\hat{x}_{1:K}$, then this finding can be applied recursively for more general (but still possibly restricted) $\mathcal{B}' \supset \mathcal{B}$.

$$
\begin{aligned}
\text{Var}_{x_{1:K} \sim q_{\mathcal{B}}} \left[ \omega_{\mathcal{B}}(x_{1:K}) \right] &= \mathbb{E}_{x_{1:K} \sim q_{\mathcal{B}}} \left[ (\omega_{\mathcal{B}}(x_{1:K}) - \bar{\omega}_{\mathcal{B}})^2 \right] \\
&= \mathbb{E}_{x_{1:K} \sim q_{\mathcal{B}}} \left[ \omega_{\mathcal{B}}(x_{1:K})^2 \right] - \bar{\omega}_{\mathcal{B}}^2 \\
&= \sum_{x_{1:K} \in \mathcal{Q} \setminus \mathcal{B}} q_{\mathcal{B}}(x_{1:K}) \omega_{\mathcal{B}}(x_{1:K})^2 - \bar{\omega}_{\mathcal{B}}^2
\end{aligned}
$$

It then follows that

$$
\begin{aligned}
\Delta_{\text{Var}} &= \bar{\omega}_{\mathcal{B}'}^2 - \bar{\omega}_{\mathcal{B}}^2 + \sum_{x_{1:K} \in \mathcal{Q} \setminus \mathcal{B}} q_{\mathcal{B}}(x_{1:K}) \omega_{\mathcal{B}}(x_{1:K})^2 - \sum_{x_{1:K} \in \mathcal{Q} \setminus \mathcal{B}'} q_{\mathcal{B}'}(x_{1:K}) \omega_{\mathcal{B}'}(x_{1:K})^2 \\
&= (\bar{\omega}_{\mathcal{B}'}^2 - \bar{\omega}_{\mathcal{B}}^2) + q_{\mathcal{B}}(\hat{x}_{1:K}) \omega_{\mathcal{B}}(\hat{x}_{1:K})^2 + \\
&\qquad \sum_{x_{1:K} \in \mathcal{Q} \setminus \mathcal{B}'} \left[ q_{\mathcal{B}}(x_{1:K}) \omega_{\mathcal{B}}(x_{1:K})^2 - q_{\mathcal{B}'}(x_{1:K}) \omega_{\mathcal{B}'}(x_{1:K})^2 \right]
\end{aligned} \tag{12}
$$

We will now analyze each of the three terms in Eq. (12) to determine when $\Delta_{\text{Var}} \geq 0$. For the first term, it follows that:

$$\bar{\omega}_{\mathcal{B}'} + \bar{\omega}_{\mathcal{B}} = p_\theta^*(X_{1:K} \in \mathcal{Q} \setminus \mathcal{B}') + p_\theta^*(X_{1:K} \in \mathcal{Q} \setminus \mathcal{B})$$
$$= 2p_\theta^*(X_{1:K} \in \mathcal{Q} \setminus \mathcal{B}') + p_\theta^*(\hat{x}_{1:K})$$
$$\bar{\omega}_{\mathcal{B}'} - \bar{\omega}_{\mathcal{B}} = p_\theta^*(X_{1:K} \in \mathcal{Q} \setminus \mathcal{B}') - p_\theta^*(X_{1:K} \in \mathcal{Q} \setminus \mathcal{B})$$
$$= -p_\theta^*(\hat{x}_{1:K})$$
$$\implies \bar{\omega}_{\mathcal{B}'}^2 - \bar{\omega}_{\mathcal{B}}^2 = -p_\theta^*(\hat{x}_{1:K})\left(2p_\theta^*(X_{1:K} \in \mathcal{Q} \setminus \mathcal{B}') - p_\theta^*(\hat{x}_{1:K})\right) \tag{13}$$

The other two terms in Eq. (12) must sum to a positive value with as large or larger magnitude to Eq. (13) for $\Delta_{\text{Var}} \geq 0$. Looking at the second term, we see that

$$q_{\mathcal{B}}(\hat{x}_{1:K})\omega_{\mathcal{B}}(\hat{x}_{1:K})^2 = \frac{p_\theta^*(\hat{x}_{1:K})^2}{q_{\mathcal{B}}(\hat{x}_{1:K})}$$

$$= p_\theta^*(\hat{x}_{1:K})(1 - q(X_{1:K} \in \mathcal{B})) \prod_{k=1}^{K} p_\theta^*(X_k \in \mathcal{V}_k | X_{<k} = \hat{x}_{<k}) \geq 0.$$

This inequality becomes strict should $p_\theta^*(\hat{x}_{1:K}) > 0$. We will now look at the final summation in Eq. (12):

$$\sum_{x_{1:K} \in \mathcal{Q} \setminus \mathcal{B}'} \left[ q_{\mathcal{B}}(x_{1:K})\omega_{\mathcal{B}}(x_{1:K})^2 - q_{\mathcal{B}'}(x_{1:K})\omega_{\mathcal{B}'}(x_{1:K})^2 \right]$$

$$= \sum_{x_{1:K} \in \mathcal{Q} \setminus \mathcal{B}'} \left( \frac{p_\theta^*(x_{1:K})^2}{q_{\mathcal{B}}(x_{1:K})} - \frac{p_\theta^*(x_{1:K})^2}{q_{\mathcal{B}'}(x_{1:K})} \right)$$

$$= \sum_{x_{1:K} \in \mathcal{Q} \setminus \mathcal{B}'} p_\theta^*(x_{1:K})\left(q(X_{1:K} \in \mathcal{B}') - q(X_{1:K} \in \mathcal{B})\right) \prod_{k=1}^{K} p_\theta^*(X_k \in \mathcal{V}_k | X_{<k} = x_{<k})$$

$$= q(\hat{x}_{1:K}) \sum_{x_{1:K} \in \mathcal{Q} \setminus \mathcal{B}'} p_\theta^*(x_{1:K}) \prod_{k=1}^{K} p_\theta^*(X_k \in \mathcal{V}_k | X_{<k} = x_{<k})$$

$$= \frac{p_\theta^*(\hat{x}_{1:K})}{\prod_{k=1}^{K} p_\theta^*(X_k \in \mathcal{V}_k | X_{<k} = \hat{x}_{<k})} \sum_{x_{1:K} \in \mathcal{Q} \setminus \mathcal{B}'} p_\theta^*(x_{1:K}) \prod_{k=1}^{K} p_\theta^*(X_k \in \mathcal{V}_k | X_{<k} = x_{<k}).$$

Since all terms in Eq. (12) have a common factor $p_\theta^*(\hat{x}_{1:K})$, we can see that $\Delta_{\text{Var}} \geq 0$ iff the following holds:

$$2p_\theta^*(X_{1:K} \in \mathcal{Q} \setminus \mathcal{B}') - \frac{1}{\rho(\hat{x}_{1:K})} \sum_{x_{1:K} \in \mathcal{Q} \setminus \mathcal{B}'} p_\theta^*(x_{1:K})\rho(x_{1:K})$$
$$\leq (1 - q(X_{1:K} \in \mathcal{B}))\rho(\hat{x}_{1:K}) + p_\theta^*(\hat{x}_{1:K}) \tag{14}$$

for $\rho(x_{1:K}) = \prod_{k=1}^{K} p_\theta^*(X_k \in \mathcal{V}_k | X_{<k} = x_{<k})$. Should this hold true, then by taking $\mathcal{B}'$ instead of $\mathcal{B}$ during the beam search segment of the hybrid approach would the variance of the sampling subroutine reduce. Generalizing this further, it is not guaranteed that the hybrid estimate will have a lower variance than regular importance sampling; however, our experimental results across a variety of settings (see Fig. 3 in the main paper) seem to indicate that the variance is reduced on average.

All of the terms to the left of the inequality in Eq. (14) are quantities that would require either expansive computations or estimation in order to know their values. Conversely, all the values to the right of the inequality are readily available as a byproduct of beam search. Incorporating this into decision making for our hybrid method is left for future work.

# F   Experimental Setup and Preparation (Section 5 in paper)

In this section we disclose all dataset preparation and modeling details necessary for reproducing our experimental results.

### F.1 Datasets

We evaluate our query estimation methods on three user behavior and two language datasets. We provide details on the preparation and utilization of each below. For all datasets, users are associated with anonymous aliases to remove personally identifiable information (PII).

**Reviews** [Ni et al., 2019] contains sequences of 233 million timestamped Amazon product reviews spanning from May 1996 to October 2018, with each product belonging to one of 30 product categories. We restrict our consideration of this dataset to reviews generated by users with at least 15 product reviews and products with a defined category, retaining 63 million reviews on which the model was trained. This dataset is publicly available under the Amazon.com Conditions of Use License.

**Mobile Apps** [Aliannejadi et al., 2021] consists of 3.6 million app usage records from 200 Android users from September 2017 to May 2018, where each event is an interaction of an individual with an application. User behavior spans 87 unique applications, which we use as the vocabulary for events for our experiments. This dataset is released under the Creative Commons License, and all users contain at least 15 mobile app interactions so no data was removed before training.

**MOOCs** [Kumar et al., 2019] is a dataset of sequences of anonymized user interactions with online course materials from a set of massive open online courses (MOOCs). In total, the dataset includes 97 unique types of interactions. Data from users with fewer than 15 interaction events are not considered, resulting in a dataset of 72% of users and 93% of the events (350,000 interactions) of the original dataset. The MOOCs dataset is available under the MIT License.

**Shakespeare** We also examine character-level language models, using the complete works of William Shakespeare [Shakespeare], comprising 125,000 lines of text and 67 unique characters and released under the Project Gutenberg License.

**WikiText** The WikiText-v2 dataset [Merity et al., 2017] includes word-level language data from "verified Good" and featured articles of Wikipedia. All sentences are provided in English and the dataset is available under the Creative Commons Attribution-ShareAlike License.

### F.2 Neural Sequence Models

For all datasets except WikiText (word-level language), we trained a 2-layer LSTM with a dropout rate of 0.3 and the ReLU activation function. Each network was trained against cross entropy loss with the Adaptive Moments (Adam) optimizer initialized with a learning rate of 0.001. A constant learning rate decay schedule and 0.01 warm-up iteration percentage was also used. All LSTM models maintain a hidden state size of 512 except the model for Shakespeare, which possesses a reduced hidden state size of 128 to reflect the size of the dataset. Model checkpoints were collected for all models every 2 epochs, and the checkpoint with the highest validation accuracy was selected to be used for query estimation experiments. All models were trained on NVIDIA GeForce 2080ti GPUs.

For WikiText-v2, we leveraged the GPT-2 medium (350 million parameters) Architecture from HuggingFace with pre-trained weights provided by OpenAI. The WikiText-v2 dataset was preprocessed using the tokenization scheme provided by HuggingFace for GPT-2, assigning numeric token indices to work pieces. No finetuning of GPT-2 is conducted.

### F.3 Ground Truth Calculations

Exact computation of ground truth is intractable for queries with large path spaces. We circumvent this issue via the law of large numbers by computing **surrogate ground truth** query estimates with a large computational budget, leveraging the variance of the query's samples as a convergence criterion. Specifically, our algorithm is conducted as follows. We first specify a *minimum* number of samples $S_{\text{low}} = 10000$ to be drawn for the surrogate ground-truth estimate. Once $S_{\text{low}}$ samples have been drawn, we compute the variance of our estimate $\hat{p}_\theta^*(X_{1:K} \in \mathcal{Q}) := \frac{1}{S} \sum_{i=1}^{S} \frac{p_\theta^*(X_{1:K}=x_{1:K}^{(i)})}{q(X_{1:K}=x_{1:K}^{(i)})}$ for

$x_{1:K}^{(1)}, \ldots, x_{1:K}^{(S)} \overset{iid}{\sim} q(X_{1:K})$:

$$\widehat{\mathrm{Var}}_q\left[\hat{p}_\theta^*(X_{1:K} \in \mathcal{Q})\right] = \frac{1}{S}\sum_{i=1}^{S}\left(\frac{p_\theta^*(X_{1:K} = x_{1:K}^{(i)})}{q(X_{1:K} = x_{1:K}^{(i)})} - \hat{p}_\theta^*(X_{1:K} \in \mathcal{Q})\right)^2$$

We then evaluate $\widehat{\mathrm{Var}}_q\left[\hat{p}_\theta^*(X_{1:K} \in \mathcal{Q})\right]$ every 1000 additional samples until either it drops below tolerance $\delta = 1e^{-7}$ or $S$ meets our maximum sample budget $S_{\mathrm{high}} = 100000$. This procedure is done in all of our experiments in which a method's performance is being compared to a query's ground truth value and exact ground truth cannot be computed due to resource constraints (typically when $K > 4$).

### F.4 Model Budget Determination

Each of the estimation methods that have been discussed can use varying amounts of computation depending on their configuration. To ensure an even comparison between methods, we configure them throughout our experiments to have roughly the same amount of computation. We measure amount of computation by the number of **model calls** $f_\theta(h_k)$ used within a given method. This can be controlled directly for all of the methods except the hybrid approach. For this reason, in our experiments we typically use fix the number of samples $S$ for the importance sampling component of the hybrid method, where $S$ is the number of samples used by the hybrid method after conducting tail-splitting beam search. We then use the resulting number of total model calls used by the hybrid method (including both beam search and sampling with $S$ samples) to determine a fixed computation budget (number of model calls to compute $f_\theta(h_k)$) for all other methods. Should the hybrid method not be used in an experiment, the budget is set by determining the number of model calls used in drawing $S$ samples for importance sampling.

## G Query Estimation Process and Examples (Section 1 in paper)

Below, we outline in clear terms the process of conducting query estimation experiments that we leverage in the main paper. Furthermore, we include demonstrative examples of potential queries of interest.

**Query Estimation Process**

1. Sample a sequence $x$ of length $n$ from the test set.
2. Using a model $p_\theta$ that was trained on the training split of the dataset, condition on the first 5 elements, $x_{1:5}$.
3. Then, using the proposed method of interest (e.g., importance sampling, coverage-based beam search, etc.), approximate the query of interest for the future continuation of the sequence–typically K steps into the future, $x_{6:6+K}$. As mentioned in lines 256 and 257 of the main paper, the main experiments pertain to hitting time queries $\tau(a) = K$ where $K = 3, \ldots, 11$ and $a$ is determined as being equal to the actual value of $x_{6+K}$ from the sampled sequence. We ensure $6 + K \leq n$.
4. Repeat steps 1-3 over 1000 sequences (unless the dataset is WikiText, in which case do 100 sequences).
5. Against either $p$ determined by absolute ground truth, or pseudo-ground truth $p$ computed via sampling, compute relative absolute error (RAE) of the form $\frac{|\hat{p}-p|}{p}$. For a given dataset, query estimation method, and budget, compute the median RAE over all 1000 (or 100 in the case of WikiTest) sampled sequences, where a sampled sequence is a history-target tuple $[\mathcal{H}, a]$.

**MOOCs** Query Example:

- **History** $\mathcal{H} = [\texttt{log on, open assignment}, \texttt{watch lecture}]$
- **Query Class**: "A" before "B" query (Q4)
- **Formalism**: $p_\theta^*(\tau(A) < \tau(B)) = p_\theta^*(\tau(\texttt{purchase}) < \tau(\texttt{log off}))$

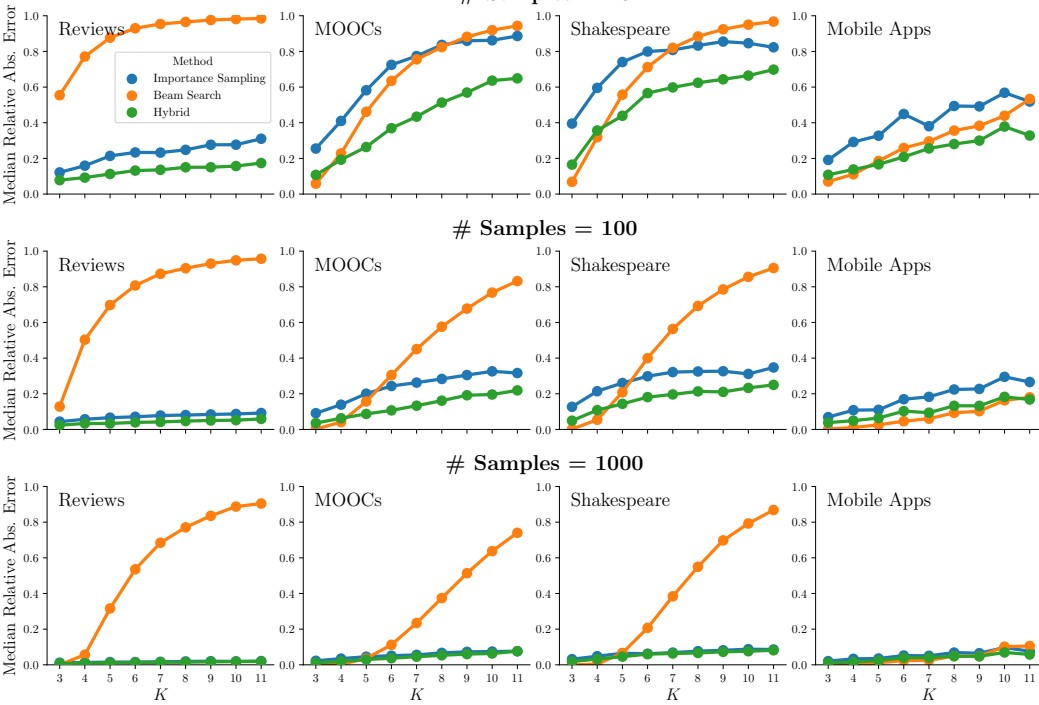

Figure 6: **Median estimation error versus query horizon** $K$: median relative absolute error (RAE) between estimated probability and (surrogate) ground truth for $p_\theta^*(\tau(\cdot) = K)$, for importance sampling, beam search, and the hybrid method, with varying computation budgets determined by 10, 100, and 1000 samples for the hybrid method. Ground truth values used to determine error in these plots are exact for $K \leq 4$ and approximated otherwise.

.

- **Description**: Given a user's online class engagement history, what is the probability that they turn in their assignment before they navigate away from the page?

**Shakespeare** Query Example:

- **History** $\mathcal{H} = $ ["t", "h", "o", "u", "<space>"]
- **Query Class**: Combination query (Q5)
- **Formalism**: $p_\theta^*(N_{\texttt{vowel}}(10) = 4)$
- **Description**: In the next 10 characters, what is the probability that 4 of them are vowels?

# H    Additional Experimental Details and Results (Section 5 in paper)

This section discusses additional experimental details and results that were not included in section 5 of our main paper due to space constraints. In all experiments, all means and medians reported are with respect to $N_Q = 1000$ randomly selected sequence locations/histories/queries per datapoint in each plot, unless stated otherwise. For each randomly selected current location, the event $a$ used in the query for $k$ steps ahead corresponds to the actual observed event $a$ for $k$ steps ahead.

## H.1    Query Estimation Error as a Function of Horizon $K$

Fig. 3 in the main paper shows the median relative absolute error (RAE) (across $N_Q = 1000$ queries) as a function of query horizon $K$ for 4 datasets, comparing beam search, importance sampling, and the hybrid method, with a computation budget fixed at $S = 100$ hybrid samples. Here we provide a number of extensions of these results.

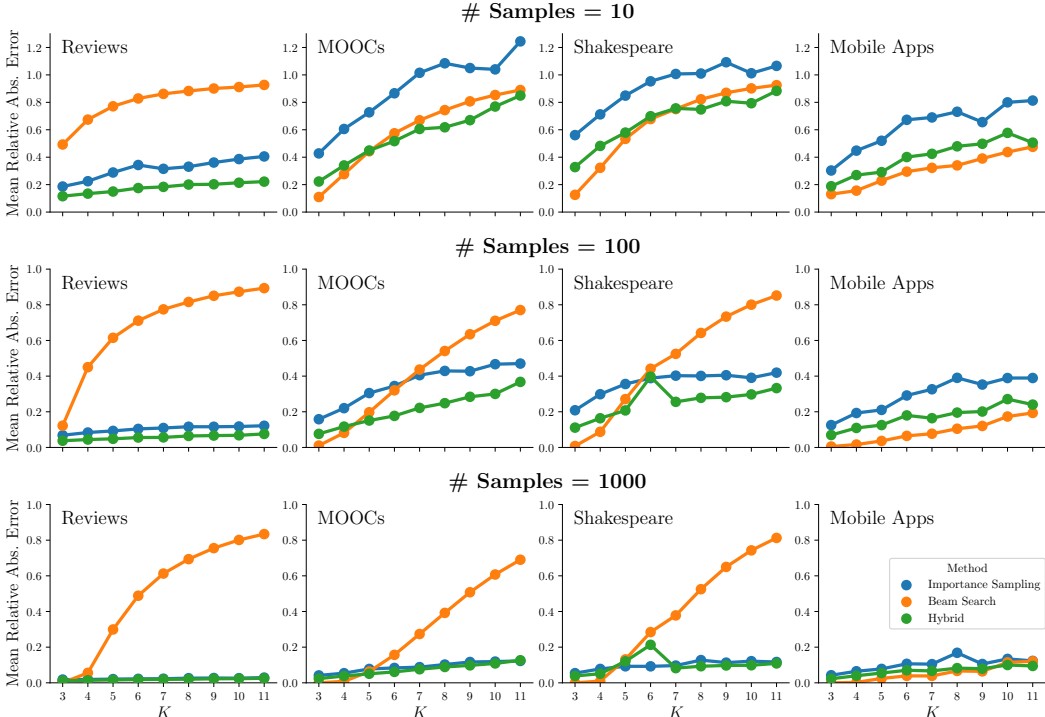

Figure 7: **Mean estimation error versus query horizon** $K$: same format as Fig. 6. Ground truth values used to determine error in these plots are exact for $K \leq 4$ and approximated otherwise.

Fig. 6 shows the median RAE, for three levels of computation budget: $S = 10, 100, 1000$ and Fig. 7 shows the same results but now reporting mean RAE on the y-axis. While the details differ across different settings, the qualitative conclusions in these Figures agree with those for Fig. 3 in the main paper, namely that beam search is more sensitive (in its error) to both the horizon query $K$ and to individual datasets, compared to both importance sampling and the hybrid method. More granular perspectives of this information for one of the datasets (MOOCs) can be seen in Figures 8 to 10 in the form of scatter plots of each of the individual query estimates against (surrogate) ground truth for that query. Different budgets are shown in different figures, and the results for $K = 3, 5, 7, 9, 11$ are shown in each column.

## H.2 Query Estimation Error as a Function of Computation Budget

In addition to identifying optimal query estimation methodologies for a low and fixed computation budget, we also explore the impact of increasing computation budget for query lengths $K = 3, 7, 11$, roughly corresponding to short, medium, and long horizon queries. These experiments are conducted in the same manner as the query estimation experiments with a fixed model budget, but are then repeated for many different budgets derived from $S = 10, 30, 50, 100, 300, 500, 1000, 3000, 5000, 10000$ hybrid samples. The intention with these experiments is to observe if any query estimation methods disproportionately benefit from increased computation and exhibit behavior that was not present at lower computation budgets. We also include two additional baselines. The full set of methods explored is listed below.

1. Importance sampling (informative proposal distribution $q$ derived from model $p_\theta$)

2. Beam search

3. Hybrid search and sampling

4. Monte-Carlo sampling with a uniform proposal distribution

5. Naive model sampling (direct MC sampling for $\mathbb{E}_{p_\theta^*} \mathbb{1}(X_{1:K} \in \mathcal{Q})$)

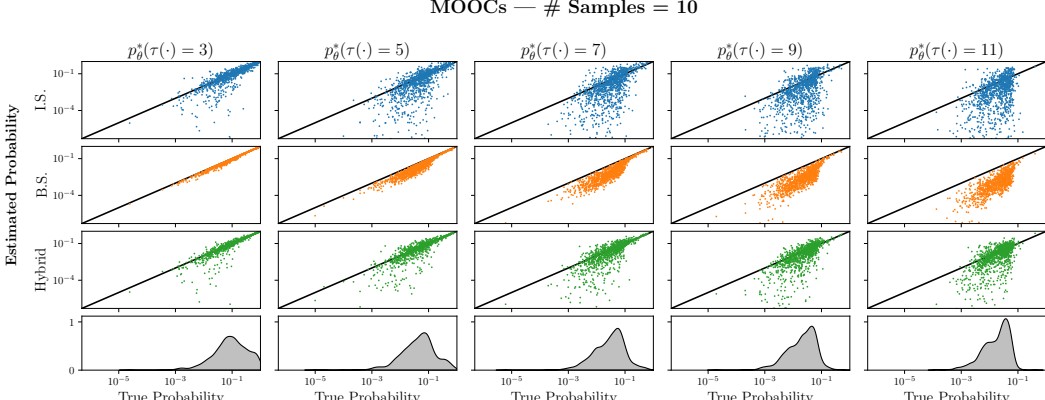

Figure 8: **Scatterplots of individual query estimates versus (surrogate) ground truth, computation budget of 10 hybrid samples:** Comparison of importance sampling (I.S.), beam search (B.S.), and the hybrid method for the MOOCs dataset with the budget determined by the hybrid method using 10 samples. The x-axis corresponds to the surrogate ground truth values for a given query result. Density plots at the bottom are for the surrogate ground truth values. Ground truth values used to determine error in these plots are exact for $K \leq 4$ and approximated otherwise.

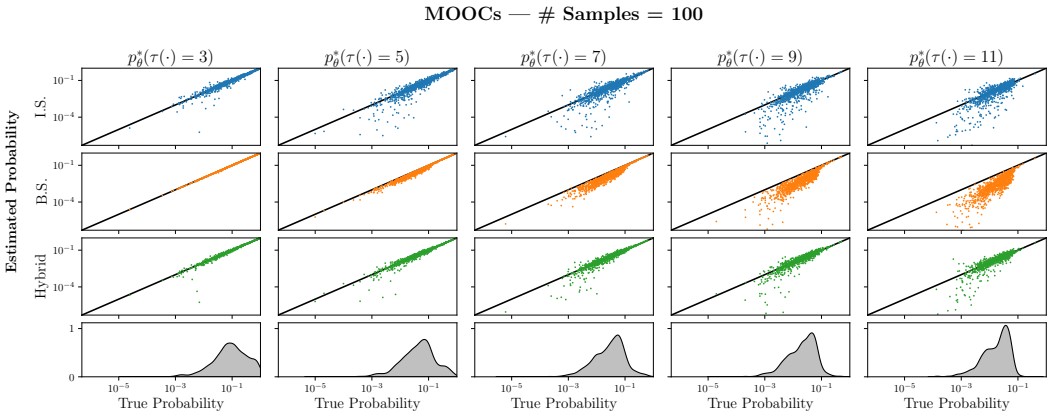

Figure 9: **Scatterplots of individual query estimates versus (surrogate) ground truth, computation budget of 100 hybrid samples:** Same format as Fig. 8.

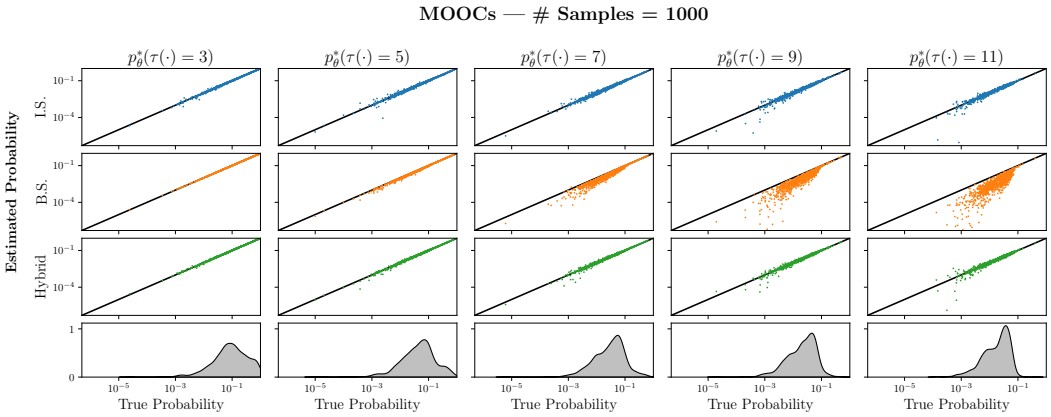

Figure 10: **Scatterplots of individual query estimates versus (surrogate) ground truth, computation budget of 1000 hybrid samples:** Same format as Fig. 8.

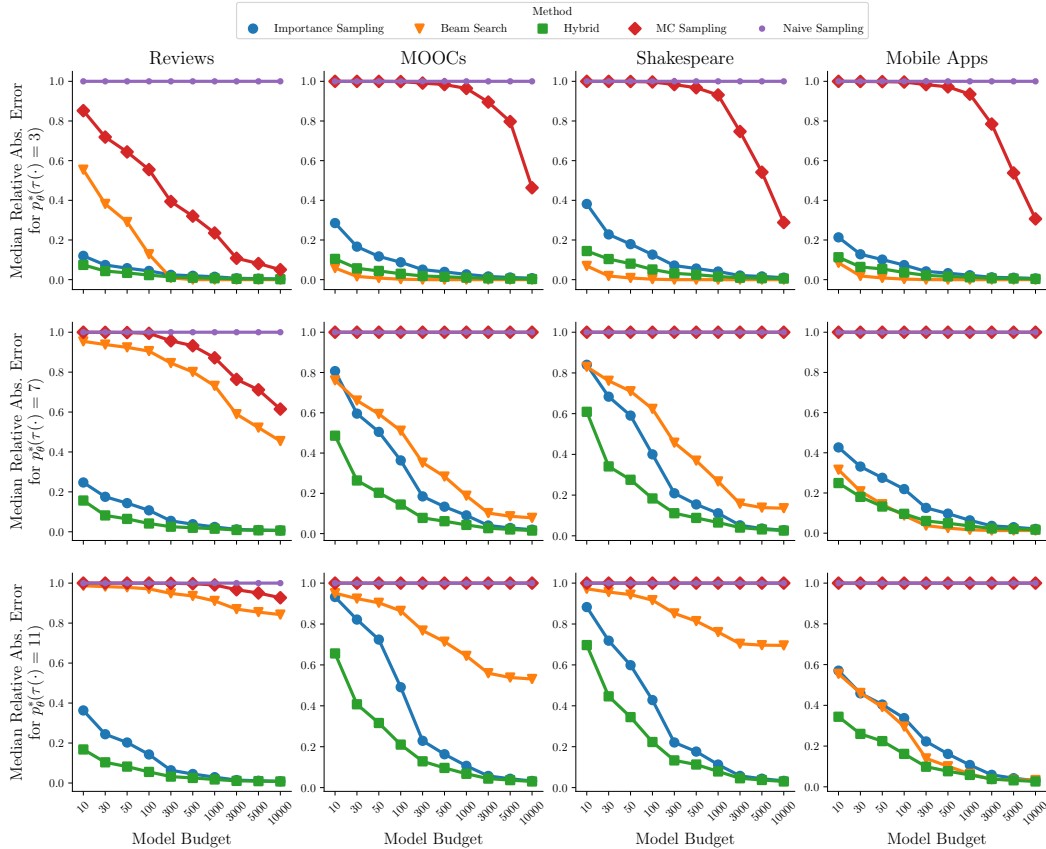

Figure 11: **Median estimation error versus model budgets:** Median relative absolute error (RAE) for importance sampling, beam search, the hybrid method, MC sampling, and naive sampling across three different queries, $p_\theta^*(\tau(\cdot) = k)$, for $k = 3, 7, 11$, over all four main datasets, as a function of different model budgets. For cases where the MC sampling results are not visible (e.g., see rightmost plot on the bottom row), the results coincide with the naive sampling results. Ground truth values used to determine error in these plots are exact for $K \leq 4$ and approximated otherwise.

As a clarifying point, naive sampling is conducted by sampling sequences from the model and determining if they fall within the query set $\mathcal{Q}$; the proportion of samples that exist in $\mathcal{Q}$ serves as a naive means of determining the query estimate in question. In addition, Monte-Carlo sampling with a uniform proposal distribution samples sequences in $\mathcal{Q}$ uniformly and the estimates the query probability for that sample. These two methods were not included in the main paper due to their consistently poor performance.

Fig. 11 (median error) and Fig. 12 (mean error), show that increasing the computation budget by an order of magnitude roughly corresponds to a three times reduction in RAE for both importance sampling and the hybrid method. Naive sampling sees almost no benefit from the increased budget regardless of the size of the query path space. Monte Carlo sampling sees some reduction in error from increased computation budget for some configurations, but also often sees no benefit.

For the provided budgets, the query estimates resulting from naive model sampling are consistently 0, resulting in an RAE of 1. While naive model sampling can be useful in some contexts in general, these results indicate that it is not well-suited for estimating query probabilities. This is likely because many of the ground truth probabilities for these queries have values on the order of $10^{-1}$ or smaller. For queries that are highly unlikely under the model, the probability of even a single sampled sequence belonging to $\mathcal{Q}$ is very low.

Monte-Carlo sampling also includes high error estimates, but for a different reason. Since the Monte-Carlo estimate can be decomposed into an expectation over $p(X_{1:K} = x_{1:K}, x_{1:K} \in \mathcal{Q})$ that

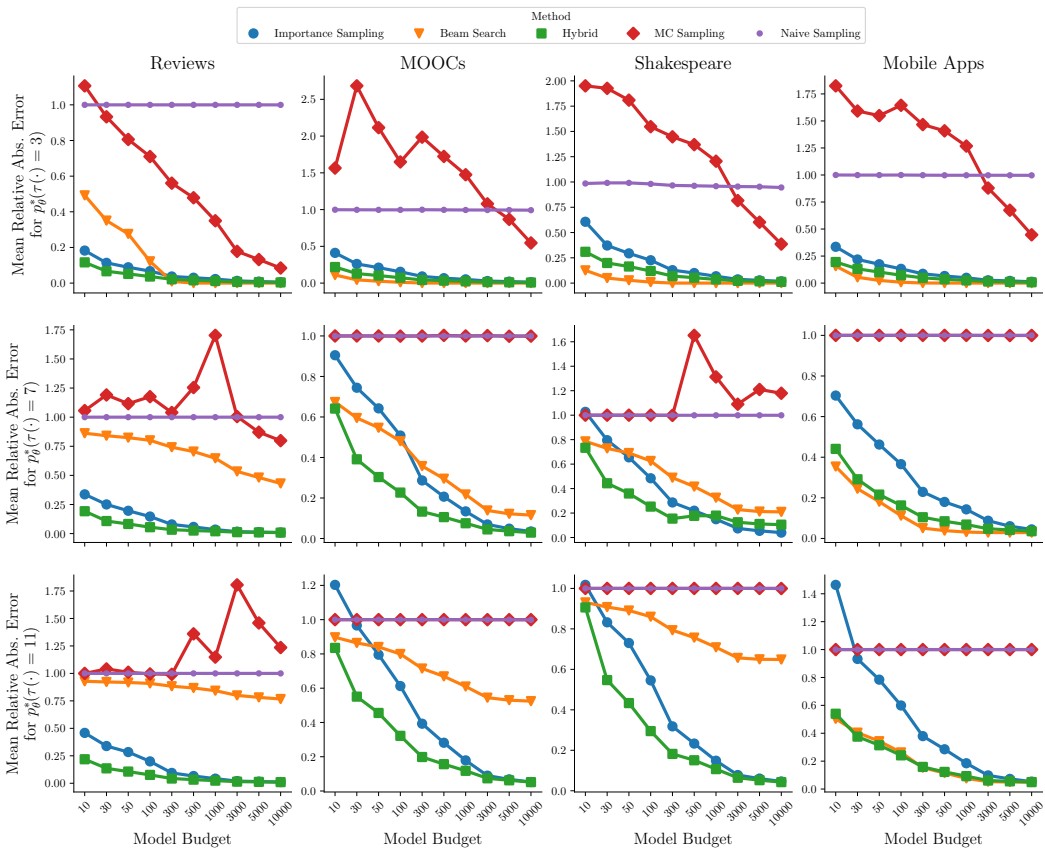

Figure 12: **Mean estimation error versus model budgets:** Same format as Fig. 11. Ground truth values used to determine error in these plots are exact for $K \leq 4$ and approximated otherwise.

is then re-scaled by $|\mathcal{Q}| \gg 0$, the scaling term magnifies any error in the expectation dramatically, inducing extremely high variance and the potential to produce query estimates that exceed 1. This high variance can persist even for high computation budgets. By contrast, beam search improves with an increased computational budget, but only as a function of the total path space. The larger the path space and the higher the entropy of the distribution, the worse the beam search estimates are as measured by RAE.

### H.3 Coverage-based Beam Search Ablation

As described in the main paper, a variant of beam search well-suited to query estimation is *coverage-based beam search*. However, this method was not explored further in our analysis as it could not scale to non-trivial query types and large query path spaces. Depicted in Fig. 13, the minimum number of beams needed to cover 50%, 75%, and 90% of the query path space increases exponentially with the query length $K$. Though coverage-based beam search comes with desirable coverage guarantees and is a consistent (albeit biased) query estimator, naively applying coverage-based beam search to queries of practical interest is ill-posed and computationally intractable. However, the hybrid method preserves some of beam search's desirable traits while making it suitable for queries of practical interest. See Appendix D for more information.

### H.4 Long-horizon Query Estimation

The empirical results of our experiments tell us much of the abilit, but we still do not have clarity on the extent to which importance sampling can yield low-error estimates in the limit of large sequence lengths. To that end, we conducted an experiment to directly assess the performance of importance sampling when estimating long horizons. Our primary conclusion was that (i) with reasonable sample

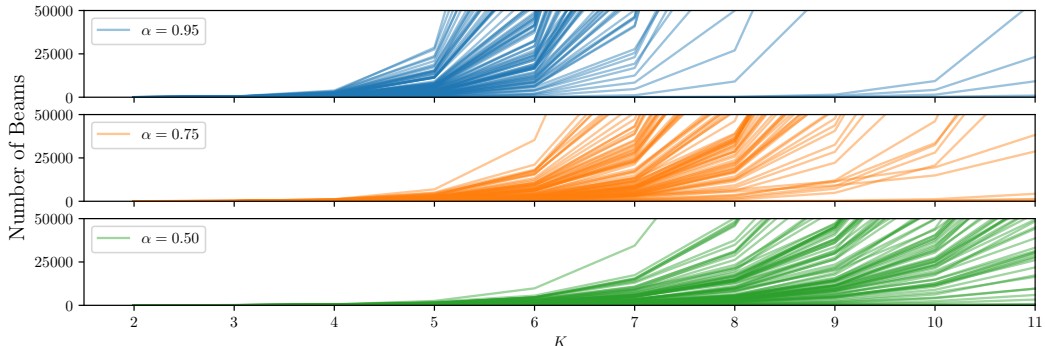

Figure 13: Number of beams as a function of $K$ steps into decoding sequences for coverage-based beam search estimating $p_\theta^*(X_K)$ with $\alpha \in \{0.95, 0.75, 0.5\}$ over a variety of different starting histories in the Shakespeare dataset.

sizes the error remains below 25%, even at $K = 100$, and (ii) the error does not increase substantially as $K$ increases beyond $K = 5$.

In more detail, a set of 100 history sequences $\mathcal{H}$ of length 5 was collected from each dataset. We then compute ground truth or pseudo ground truth (PGT) for query 2 (probability of event at $K$ without restriction) over sequence lengths $K = [2, 5, 10, 20, 40, ..., 100]$. Pseudo-ground truth estimates are generated with a tolerance $\delta = 1e^{-7}$ and a maximum sampling budget of $250,000$. The same estimates are then computed using importance sampling with sampling budgets $[10, \ldots, 10000]$. Similar to the experiments in Section 5, we aggregate and present the results as the median relative absolute error, shown in the tables below. Since we are not focusing on a specific event type of interest, this median also marginalizes over all event types in the vocabulary as well as the sampled query sequences. All results are reported as percentages

| # Samples | 2 | 5 | 10 | 20 | 40 | 60 | 80 | 100 |
|-----------|-------|-------|-------|-------|-------|-------|-------|-------|
| 10 | 31.35 | 40.33 | 44.39 | 43.48 | 45.98 | 45.84 | 49.44 | 50.05 |
| 100 | 13.29 | 17.12 | 18.73 | 19.46 | 20.41 | 21.65 | 21.82 | 21.86 |
| 1000 | 4.94 | 6.38 | 7.06 | 7.33 | 7.26 | 6.99 | 7.13 | 6.82 |
| 10000 | 1.60 | 2.05 | 2.21 | 2.34 | 2.20 | 2.13 | 2.06 | 2.10 |

Table 2: Median RAE across query estimations methods (1000 samples) and query horizons $K = 2, 5, 10, 20, ..., 100$ and sample budgets $10, 100, 1000, 10000$ for Amazon Reviews.

| # Samples | 2 | 5 | 10 | 20 | 40 | 60 | 80 | 100 |
|-----------|-------|-------|-------|-------|-------|-------|-------|-------|
| 10 | 66.21 | 84.93 | 92.92 | 96.96 | 98.41 | 99.00 | 99.19 | 99.32 |
| 100 | 62.36 | 80.94 | 89.79 | 94.00 | 96.16 | 97.05 | 97.38 | 97.66 |
| 1000 | 47.96 | 59.14 | 69.10 | 75.45 | 74.79 | 71.34 | 65.12 | 59.75 |
| 10000 | 15.21 | 20.51 | 23.38 | 24.00 | 20.66 | 18.78 | 16.48 | 15.55 |

Table 3: Median RAE across query estimations methods (1000 samples) and query horizons $K = 2, 5, 10, 20, ..., 100$ and sample budgets $10, 100, 1000, 10000$ for Mobile Apps.

In general, we see (not surprisingly) that the increase in sequence length leads to a consistent and non-trivial increase in error for most sampling budgets. In addition, as expected the increase in sampling budget consistently reduces the query estimation error. However, we do witness the interesting phenomenon that the error occasionally *decreases* as the sequence length increases. We conjecture that this may be happening because as the sequence length increases, the relevance of the history context $\mathcal{H}$ decreases and the distribution may regress to a base stationary distribution (as if no history context were provided at all) indicating that the conditional model entropy may be the main driving factor in estimation complexity. This intuition is further supported by the fact that in many datasets, budgets exist where query estimation error first increases but then begins to decrease again.

| # Samples | 2 | 5 | 10 | 20 | 40 | 60 | 80 | 100 |
|---|---|---|---|---|---|---|---|---|
| 10 | 79.84 | 79.99 | 83.53 | 85.29 | 83.53 | 84.00 | 85.82 | 85.15 |
| 100 | 28.66 | 32.85 | 39.25 | 40.66 | 38.78 | 41.51 | 39.71 | 40.07 |
| 1000 | 8.61 | 10.99 | 13.29 | 13.52 | 13.78 | 13.89 | 13.64 | 14.10 |
| 10000 | 2.77 | 3.48 | 4.28 | 4.34 | 4.32 | 4.33 | 4.34 | 4.30 |

Table 4: Median RAE across query estimations methods (1000 samples) and query horizons $K = 2, 5, 10, 20, ..., 100$ and sample budgets 10, 100, 1000, 10000 for Shakespeare.

| # Samples | 2 | 5 | 10 | 20 | 40 | 60 | 80 | 100 |
|---|---|---|---|---|---|---|---|---|
| 10 | 89.62 | 97.30 | 98.73 | 99.19 | 99.08 | 99.06 | 98.91 | 98.87 |
| 100 | 63.80 | 83.34 | 84.77 | 79.15 | 66.89 | 62.99 | 60.28 | 61.30 |
| 1000 | 30.39 | 43.37 | 36.32 | 26.07 | 19.78 | 18.11 | 17.53 | 17.38 |
| 10000 | 11.92 | 15.15 | 11.35 | 7.86 | 5.90 | 5.66 | 5.43 | 5.40 |

Table 5: Median RAE across query estimations methods (1000 samples) and query horizons $K = 2, 5, 10, 20, ..., 100$ and sample budgets 10, 100, 1000, 10000 for MOOCs.

Regardless, as the largest budget of $10,000$ we witness that median RAE remains at or under 25% in all cases, often significantly so.

## H.5  Query Estimation with Large-Scale Language Models

In order to explore the feasibility of applying our query estimation methods to real-world sequence data, we also analyze a subset of our query estimation methods against GPT-2 and WikiText language data. GPT-2 decomposes English words into $V = 50257$ work pieces, a vocabulary over 500 times larger than our other datasets. For this reason, we only conduct experiments using only 100 sequence histories per dataset due to computational limitations. For the same reason, we do not explore the hybrid method and restrict ourselves to analyzing the following query estimators:

1. Importance sampling (informative proposal distribution $q$ derived from model $p_\theta$)

2. Beam search

3. Monte-Carlo sampling with a uniform proposal distribution

Fixed-budget query experiments with GPT-2 are conducted identically to those on the other 4 datasets. We find query estimation error closely mirrors the results we see in datasets with smaller vocabulary sizes, suggesting our findings may generalize well to practical domains. Our analysis is reported in Table 6 and includes estimates of the restricted model entropy $H(q)$ for different query lengths $K$, with the entropy increasing much faster than small-vocabulary models, as expected. With that said, there is still much exploration to be done on large-scale sequence models and is a promising avenue for future work.

| K | Importance Samp. | Beam Search | MC Samp. | Entropy | Entropy % |
|---|---|---|---|---|---|
| 3 | **11.41** | 51.25 | 99.36 | 12.89 | 41.28 |
| 4 | **13.35** | 82.42 | 99.95 | 19.09 | 40.74 |
| 5 | **13.53** | 93.59 | 99.99 | 25.23 | 40.38 |

Table 6: **Median RAE across query estimations methods (1000 samples) and query horizons** $K = 3, 4, 5$ **for GPT-2 and Wikitext**. Entropy values estimated as the mean (over 100 queries) of the restricted proposal $q$ and entropy % is the entropy in percentage relative to its potential maximum value $K log(V)$.

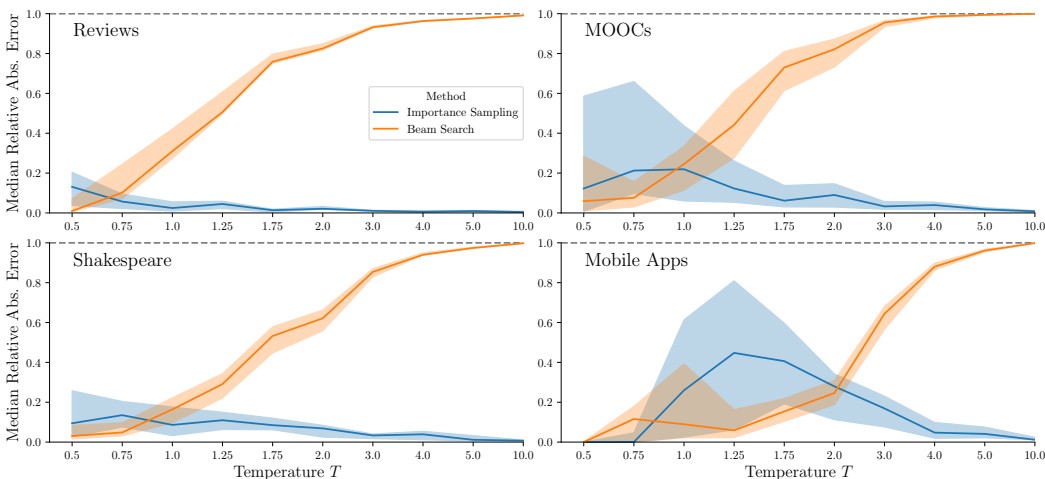

Figure 14: **Error as a function of entropy:** Median relative absolute error for estimating $p^*_{\theta,T}(\tau(\cdot) = 4)$ over all four datasets for a variety of imposed temperatures $T$. Shaded regions indicate interquartile ranges.

### H.6 Entropy Relationship with Query Estimation Error

Fig. 4(b) in the main paper demonstrated how indirectly controlling the entropy of a given model through an applied temperature affected the performance of beam search and importance sampling. In Fig. 14 we can see similar plots for all of four of our main datasets.

### H.7 Investigation of Query 4 ("A" before "B")

Most of our analysis was conducted on hitting time queries, and justifiably so as more advanced queries decompose into individual operations on hitting times. This includes Q4, colloquially stated as the probability an item from token set $A$ occurs before an item in token set $B$. More formally:

$$p^*_\theta(\tau(A) < \tau(B)) = \sum_{k=1}^{\infty} p^*_\theta(\tau(A) = k, \tau(B) > k)$$

$$= \sum_{k=1}^{\infty} \sum_{a \in A} p^*_\theta(X_k = a, X_{<k} \in (\mathbb{V} \setminus (A \cup B))^{k-1})$$

While this cannot be computed exactly, a lower bound can. The other option is to produce a lower bound on this expression by evaluating the sum in Eq. (11) for the first $K$ terms. We can achieve error bounds on this estimate by noting that $p^*_\theta(\tau(a) < \tau(b)) + p^*_\theta(\tau(a) > \tau(b)) = 1$. As such, if we evaluate Eq. (11) up to $K$ terms for both $p^*_\theta(\tau(a) < \tau(b))$ and $p^*_\theta(\tau(b) < \tau(a))$, the difference between the sums will be the maximum error either lower bound can have. This difference will be referred to as *unaccounted probability* and will approach 0 as $K \to \infty$.

A natural question to ask is what is the minimum value of $K$ sufficient to compute these lower bounds to in order to have negligible unaccounted probability. Though this will surely vary based on the entropy of the model and the specific query in question, we explore this question across all datasets except WikiText and note some general trends.

Fig. 15 plots the unaccounted probability $1 - \hat{p}^*_\theta(\tau(A) < \tau(B)) - \hat{p}^*_\theta(\tau(A) > \tau(B))$ as a function of query length $K$. We observe that for many datasets, a query horizon of $k = 30$ is largely sufficient to reduce the remaining probability to under 10%. One notable exception is the Mobile Apps dataset, which, due to its lower entropy and high self-transition rate, maintains a much longer query horizon. This discovery implies that a successful partition of probability space with a given Q4 query can be sensitive to the model distribution, but also that approximate partitions are possible for relatively low values of $K$.

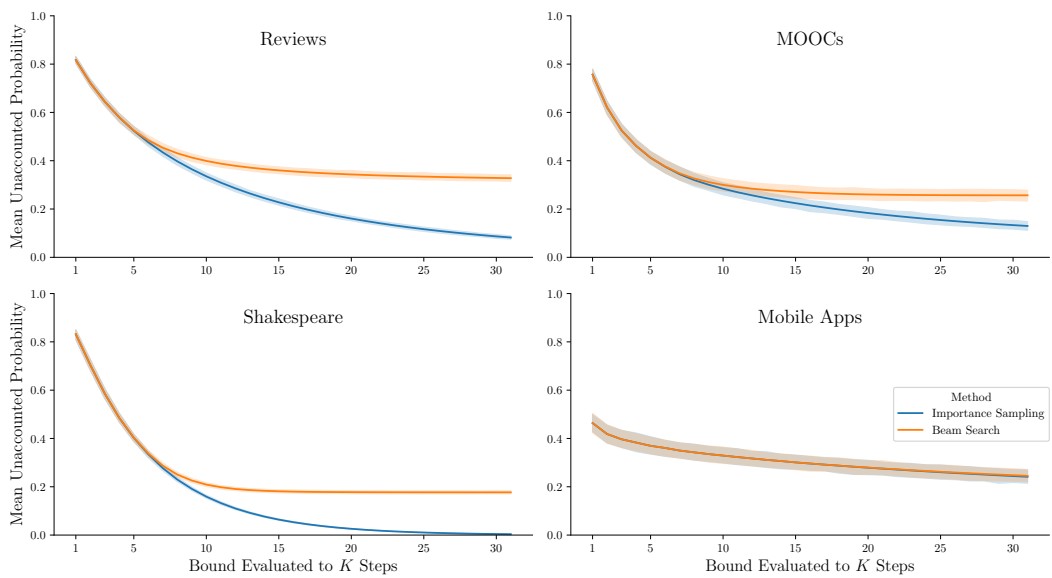

Figure 15: Mean unaccounted probability $(1 - (\hat{p}_\theta^*(\tau(A) < \tau(B)) + \hat{p}_\theta^*(\tau(B) < \tau(A))))$ when evaluating the pair of queries for $\tau(A) < \tau(B)$ and $\tau(B) < \tau(A)$ up to $K$ steps into the future over all four main datasets. Shaded regions indicate 99% confidence intervals and estimates were computed with a fixed model budget based on 1000 samples.

## H.8 Linearly Compounding Errors in Complex Query Estimation

As mentioned in Section 5 of the main paper, hitting time queries can often be seen as components of more involved queries, such as "a" before "b" queries $p_\theta^*(\tau(a) < \tau(b))$ or counting-style queries $p_\theta^*(N_a(K) = n)$. These queries can be rewritten as summations of more basic hitting time queries. For instance, $p_\theta^*(\tau(a) < \tau(b)) = \sum_{k=1}^\infty p_\theta^*(\tau(a) = k, \tau(b) > k)$. In practice, each summand probability is estimated using our proposed techniques so it can be seen that the error compounds additively with respect to the different basic hitting time queries.

More generally, our framework proposes representing general queries as $\mathcal{Q} = \cup_i \mathcal{Q}_i = \cup_i \prod_j \mathcal{V}_j^{(i)}$ such that the $\mathcal{Q}_i$'s form a minimal partition on $\mathcal{Q}$. With this representation, we can see that for an arbitrary query, the error when estimating will compound additively and scale linearly with respect to the number of different $\mathcal{Q}_i$. Note that the actual values of $p_\theta^*(x \in \mathcal{Q}_i)$ do have an impact on the errors when estimating due to the values $\in [0, 1]$. Lastly, as mentioned in the paper we can additionally control this error either by utilizing the coverage-based beam search, or by leveraging the Central Limit Theorem with importance sampling.

## H.9 Qualitative Exploration and Practical Applications

In addition to systematic quantitative analysis of query estimators, we also qualitatively explore specific applications of our methods that lend practical insights. First, we consider the question of "given a partial sentence, predict when the sentence will end". Using our query estimation methods, we can not only answer this question with relatively low error but also effectively re-use intermediate computational results. This necessarily correlates query estimates over steps $K$, but this confers little negative impact upon the analysis since our sampling methods are unbiased estimators and beam search, though biased, offers a deterministic lower bound. The results of our analysis are seen in Fig. 1 in the main paper and align with basic intuition about English sentences, with open ended prefixes possessing a long-tailed end-of-sentence probability distribution relative to more structured and declarative phrases.

A second practical question that can be asked with our query estimation methods utilizes the Mobile Apps dataset: given someone's mobile usage history, predict what will occur first: the individual will go on social media or directly interact with someone via video chat, call, text, or email. This question

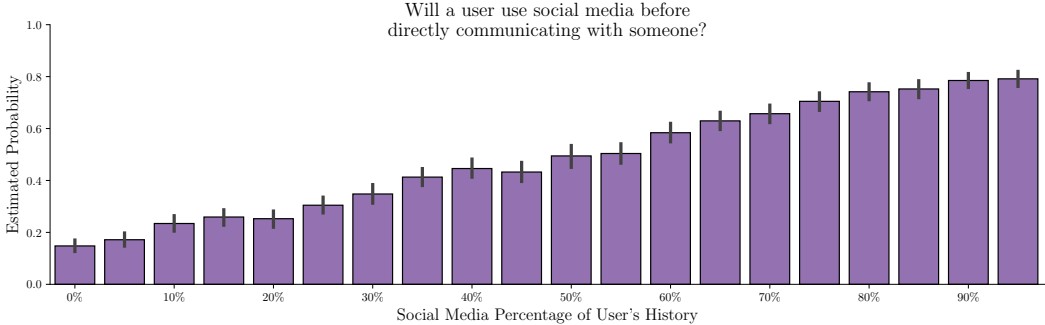

Figure 16: A case study of evaluating how likely it is that a mobile app user will use social media before they directly communicate with someone via text messaging, email, phone call, etc. as a function of how much of their recent history consists of social media usage. This is a $p_\theta^*(\tau(A) < \tau(B))$ type query that has been estimated and averaged over 500 different histories for each social media history percentage. Error bars indicate 90% confidence intervals.

is a practical application of query 4, $p_\theta^*(\tau(a) < \tau(b))$. Other, equally interesting equivalents of this question include "will an online shopper purchase something before leaving the website?". Below, we have conducted an experiment where we synthetically generate mobile app behavior histories with specific percentages of social media activity present. By computing these queries over several histories and then averaging the estimates, we obtain the results seen in Fig. 16. As expected, we see a clear linear trend between a user's social media usage and the likelihood they will return to it before conducting other tasks like directly communication. Such a result, though contrived and purely demonstrative in this setting, could be applied to many practical applications that analyze the characteristics of online user behavior.