# OpenReview forum: "Predictive Querying for Autoregressive Neural Sequence Models"
_NeurIPS.cc/2022/Conference — NeurIPS 2022 Accept_

### Official Review · Reviewer_6TXq · 2022-07-09

**Rating:** 8
**Confidence:** 4
**Soundness:** 4 excellent
**Presentation:** 3 good
**Contribution:** 4 excellent

**Summary:**

The paper introduces the problem of predictive querying in neural autoregressive models, where we answer probabilistic queries such as hitting time of an event wrt the model (not some ground truth). They propose an importance-sampling based framework and proposal distribution for estimating these queries, and also combine it with beam search for better results. The provide some understanding of the failures of beam search for predictive querying, including the correlation of beam search's error with the conditional entropy of the model.

**Questions:**

- The paper discusses coverage-based and tail splitting beam search (and perhaps claims it as a contribution?) but doesn't show its results, saying that it did not do well. Perhaps this is an extraneous part of the paper?
- While hitting time is a component of other queries, it could be interesting to see how the error compounds with multiple related queries.
- One limitation is that the evaluation method requires an expensive computation of the true probabilities. Is there a different way to verify the performance of the query estimation on longer horizons? While it's clear that the IS approach is better than beam search, it's unclear how far it can go. One possibility is to sample sequences from the models where an event actually occurs, truncating this sequence K steps before the event occurs, and then evaluating the likelihood that the event occurs under the predictive query from the past (and average over many sequences).

**Limitations:**

They discuss the potential biases of using forecasting models for decision making, especially on data like student interactions as used in the paper. They also preprocess the datasets to not contain personally identifiable information.

**Strengths And Weaknesses:**

- The paper raises an important question about how to answer probabilistic queries about future steps, which could be applied to decision making systems that require some forecasting. They also provide a framework, baselines, and an improved hybrid method, as well as some understanding of why beam search approaches are worse. The paper is well written overall, though the explanation/motivation of the proposal distribution could be made clearer.
- On originality, while the setup seems new for neural autoregressive models, the family of problems is well studied in the stochastic processes literature, survival analysis, etc, but as far as I know they usually assume some parametric form for the underlying probabilistic model / hazard function / etc. that is easier to work with. For neural models there is less structure to work with.
- The paper shows a nice opposite behavior for beam search and Importance sampling methods, and proposes a hybrid approach that nicely combines the two for mutual benefits, using importance sampling to estimate the gap between the beam search lower bound and the true probability.
- The paper considers a variety of sequences (text, interactions, usage records) and models (LSTMs, GPT2 (which one?) ).

---

> ### Author Response · Authors · 2022-08-01
> **Response to Reviewer 6TXq (PART 1/2)**
>
> We greatly appreciate the enthusiasm and interest the reviewer has shown towards our work. We thank the reviewer for their comments and questions and have responded to each below.
>
> > … the explanation/motivation of the proposal distribution could be made clearer.
>
> We thank the reviewer for pointing this out. In the final version of the paper we will clarify the discussion of our proposal distribution in section 4.1 of our main paper to better connect our motivation with the query estimation methodologies of section 4.2.
>
> > (GPT2 version?)
>
> We leverage the GPT2 medium architecture hosted on HuggingFace with pre-trained weights provided by OpenAI. We will add this information to the paper.
>
> > [Q1] (Coverage-based and tail-splitting beam search)
>
> We appreciate the reviewer for bringing this up as it appears our text may not have been clear regarding this topic. We present both coverage-based and tail-splitting beam search as contributions; however, they serve widely different purposes. The coverage-based beam search was created when it is desired to calculate a query to a known percent coverage of the search space (proportional to the model). For queries with very large search spaces though, this can be quite a problem and quickly lead to out-of-memory issues. This behavior can be seen in experiments we performed in the Appendix (see Section F.3). The tail-splitting beam search, however, is used throughout our other experiments, including all of Section 5. This variant was developed as a compromise between setting an arbitrary fixed number of beams and having a target coverage rate of the search space. In practice, this method is able to go out decently far into the future for sequences and not run into immediate memory issues. We will be sure to make the discussion of how we use these methods clearer in the paper.
>
> > [Q2] (Compounding errors of hitting time)
>
> We appreciate the reviewer’s suggestion! As mentioned, hitting time queries can often be seen as components of more involved queries, such as “a” before “b” queries $p_\theta^*(\tau(a) < \tau(b))$ or counting-style queries $p_\theta^*(N_a(K)=n)$. These queries can be rewritten as summations of more basic hitting time queries. For instance, $p_\theta^*(\tau(a) < \tau(b)) = \sum_{k=1}^\infty p_\theta^*(\tau(a)=k,  \tau(b)>k)$. In practice, each summand probability is estimated using our proposed techniques so it can be seen that the error compounds additively with respect to the different basic hitting time queries.
>
> More generally, our framework proposes representing general queries as $\mathcal{Q} = \cup_i \mathcal{Q}_i = \cup_i \prod_j \mathcal{V}_j^{(i)}$ such that the $\mathcal{Q}_i$’s form a minimal partition on $\mathcal{Q}$. With this representation, we can see that for an arbitrary query, the error when estimating will compound additively and scale linearly with respect to the number of different $\mathcal{Q}_i$. Note that the actual values of $p_\theta^*(x\in\mathcal{Q}_i)$ do have an impact on the errors when estimating due to the values $\in[0,1]$. Lastly, as mentioned in the paper we can additionally control this error either by utilizing the coverage-based beam search, or by leveraging the Central Limit Theorem with importance sampling.
>
> We see this as an interesting topic for future research and we will mention these points in the Discussion section of the paper.
>
> > [Q3.1] (Unclear how far IS can go)
>
> We thank the reviewer for this excellent question! We agree that we do not have clarity on the extent to which importance sampling can yield low-error estimates in the limit of large sequence lengths. To that end, we conducted an experiment during the author response period to directly assess the performance of importance sampling when estimating long horizons. Our primary conclusion was that (i) with reasonable sample sizes the error remains below 25%, even at $K=100$, and (ii) the error does not increase substantially as $K$ increases beyond $K=5$. We will be adding the experimental results and discussions in the Appendix.
>
> In more detail, a set of 100 history sequences $\mathcal{H}$ of length 5 was collected from each dataset. We then compute ground truth or pseudo ground truth (PGT) for query 2 (probability of event at $K$ without restriction) over sequence lengths $K =[2, 5, 10, 20, 40, ..., 100]$. Pseudo-ground truth estimates are generated with a tolerance $\delta=1e^{-7}$ and a maximum sampling budget of $250,000$. The same estimates are then computed using importance sampling with sampling budgets $[10, …, 10000]$. Similar to the experiments in Section 5, we aggregate and present the results as the median relative absolute error, shown in the tables below. Since we are not focusing on a specific event type  of interest, this median also marginalizes over all event types  in the vocabulary as well as the sampled query sequences. All results are reported as percentages.
>
> _Please refer to Part 2/2 for results and further comments._

---

> > ### Author Response · Authors · 2022-08-01
> > **Response to Reviewer 6TXq (PART 2/2)**
> >
> > Below are the tables of results for the experiment described previously. Columns refer to median relative absolute errors at various steps into the future for $K\in\\{2, 5, 10, 20, 40, 60, 80, 100\\}$. Rows refer to results across different sampling budgets used when producing estimates.
> >
> > **Amazon Reviews (Vocab size $V=29$)**
> >
> > | \# Samples | 2     | 5     | 10    | 20    | 40    | 60    | 80    | 100   |
> > |------------|-------|-------|-------|-------|-------|-------|-------|-------|
> > | 10         | 31.35 | 40.33 | 44.39 | 43.48 | 45.98 | 45.84 | 49.44 | 50.05 |
> > | 100        | 13.29 | 17.12 | 18.73 | 19.46 | 20.41 | 21.65 | 21.82 | 21.86 |
> > | 1000       | 4.94  | 6.38  | 7.06  | 7.33  | 7.26  | 6.99  | 7.13  | 6.82  |
> > | 10000      | 1.60  | 2.05  | 2.21  | 2.34  | 2.20  | 2.13  | 2.06  | 2.10  |
> >
> > **Apps (Vocab size $V=88$)**
> >
> > | \# Samples | 2     | 5     | 10    | 20    | 40    | 60    | 80    | 100   |
> > |------------|-------|-------|-------|-------|-------|-------|-------|-------|
> > | 10         | 66.21 | 84.93 | 92.92 | 96.96 | 98.41 | 99.00 | 99.19 | 99.32 |
> > | 100        | 62.36 | 80.94 | 89.79 | 94.00 | 96.16 | 97.05 | 97.38 | 97.66 |
> > | 1000       | 47.96 | 59.14 | 69.10 | 75.45 | 74.79 | 71.34 | 65.12 | 59.75 |
> > | 10000      | 15.21 | 20.51 | 23.38 | 24.00 | 20.66 | 18.78 | 16.48 | 15.55 |
> >
> >
> > **Shakespeare (Vocab size $V=67$)**
> > | \# Samples | 2     | 5     | 10    | 20    | 40    | 60    | 80    | 100   |
> > |------------|-------|-------|-------|-------|-------|-------|-------|-------|
> > | 10         | 79.84 | 79.99 | 83.53 | 85.29 | 83.53 | 84.00 | 85.82 | 85.15 |
> > | 100        | 28.66 | 32.85 | 39.25 | 40.66 | 38.78 | 41.51 | 39.71 | 40.07 |
> > | 1000       | 8.61  | 10.99 | 13.29 | 13.52 | 13.78 | 13.89 | 13.64 | 14.10 |
> > | 10000      | 2.77  | 3.48  | 4.28  | 4.34  | 4.32  | 4.33  | 4.34  | 4.30  |
> >
> > **MOOCs (Vocab size $V=98$)**
> > | \# Samples | 2     | 5     | 10    | 20    | 40    | 60    | 80    | 100   |
> > |------------|-------|-------|-------|-------|-------|-------|-------|-------|
> > | 10         | 89.62 | 97.30 | 98.73 | 99.19 | 99.08 | 99.06 | 98.91 | 98.87 |
> > | 100        | 63.80 | 83.34 | 84.77 | 79.15 | 66.89 | 62.99 | 60.28 | 61.30 |
> > | 1000       | 30.39 | 43.37 | 36.32 | 26.07 | 19.78 | 18.11 | 17.53 | 17.38 |
> > | 10000      | 11.92 | 15.15 | 11.35 | 7.86  | 5.90  | 5.66  | 5.43  | 5.40  |
> >
> > In general, we see (not surprisingly) that the increase in sequence length leads to a consistent and non-trivial increase in error for most sampling budgets. In addition, as expected the increase in sampling budget consistently reduces the query estimation error. However, we do witness the interesting phenomenon that the error occasionally _decreases_ as the sequence length increases. We conjecture that this may be happening because as the sequence length increases, the relevance of the history context $\mathcal{H}$ decreases and the distribution may regress to a base stationary distribution (as if no history context were provided at all) indicating that the conditional model entropy may be the main driving factor in estimation complexity. This intuition is further supported by the fact that in many datasets, budgets exist where query estimation error first increases but then begins to decrease again. Regardless, as the largest budget of $10,000$ we witness that median RAE remains at or under 25% in all cases, often significantly so.
> >
> >
> >
> > > [Q3.2] … One possibility is to sample sequences from the models …
> >
> > This is an interesting suggestion. Using this idea we could straightforwardly generate samples that satisfy a predetermined query of interest by utilizing our novel proposal distribution. However, we fear that there may be issues in evaluating the performance of an approximation method purely by observing the likelihood as proposed. Low likelihoods for a particular history and query are not necessarily indicative of a poor estimation as the ground truth probability for the query could naturally be small. Possibly the only guarantee one may have is that, if done over many different truncated sample sequences and queries, likelihoods will be higher on average when the query matches the generated sequence than when they do not match. This unfortunately does not necessarily speak to the precision / accuracy of the approximation method.
> >
> > If we have misinterpreted your suggestion please do let us know. This line of evaluation is very important for future work as it greatly resembles what we believe to be a promising approach to evaluating predictive queries on _real world data_ rather than against a model’s beliefs.

---

> > > ### Comment · Reviewer_6TXq · 2022-08-04
> > > **Thanks for the response**
> > >
> > > Thanks for the response and for running additional experiments! The phenomenon where the errors can improve with length is quite interesting and deserves a bit more digging into. It does seem that evaluation for this problem is tricky and important to figure out - we would of course eventually want to get to a point where we are using these models in a useful regime (where we cannot brute force the probability estimate).

---

### Official Review · Reviewer_yqp6 · 2022-07-10

**Rating:** 6
**Confidence:** 3
**Soundness:** 4 excellent
**Presentation:** 4 excellent
**Contribution:** 3 good

**Summary:**

This paper proposes a new problem for neural autoregressive sequence models: How do we query the model about probabilistic questions like "what is the probability of A occurring before B". Given a pretrained sequence model, it is usually intractable to query the model to answer those questions, so the paper proposes 2 estimation methods: importance sampling and beam search. The paper also tries to combine those 2 methods and finds that the combination of them achieves the best results.

**Questions:**

1. Given the sequence space can be large and the importance sampling has non-negligible variances, it is helpful to show the effective sampling size.
2. How do you think of ISR (importance sampling resampling)? Resampling can significantly improve the coverage of the importance sampling and is unbiased, which can be a surrogate of the beam search.
3. How do you balance importance sampling and beam search in the hybrid method? I wonder about the ratio of their sequences and possibly their contributions to the resulting probability (i.e. L202).

**Limitations:**

The societal impacts of this work are already discussed by the authors and I don't have further concerns.

**Strengths And Weaknesses:**

The problem raised by this paper is interesting and was thought intractable before. The problem of token counting or the ordering of some tokens can be potentially useful for many real applications, e.g. finance and NLP.

Although the methods in this paper are existing ones, certain tricks are applied to each of them, for example, coverage-based beam-search and tail-splitting search. The hybrid method combines the advantage of sampling and beam search, which yields the best results.

The paper is well-motivated and well-written and can be valuable for many other fields. However, it does come with certain drawbacks:

1. The 2 methods used in the paper are existing ones and lack novelty. Applying them to the estimation of sequences is natural but less non-trivial.
2. The definition of "probabilistic queries" is big and only parts of them are discussed in the paper. When the Q space gets larger, the estimation gets harder under the framework of this work. For example, the Q5 cannot be easily handled when C(K, n) gets large. In fact, only hitting time (Q3) is tested in section 5.
3. Furthermore, the experiments are not sufficient and the actual metric relies on the sampling (L266). No baselines are available, which makes it hard to judge the contribution of this paper. A suggestion would be training a supervised model for each K to predict the hitting time.
4. I have a few methodological concerns that are listed in the question section.

---

> ### Author Response · Authors · 2022-08-01
> **Response to Reviewer yqp6 (PART 1/2)**
>
> We appreciate the in-depth questions and suggestions given by the reviewer. We have reproduced them and responded to each below.
>
> > [W1] ....lack novelty…
>
> While we agree that importance sampling and beam search are existing methods, we argue that the novel contribution in our work is rather the following:
> 1. The general formulation of the “probabilistic query” framework and representation (refer to Section 3 and Equations 1-3).
> 2. The tractable proposal distribution (Equations 5 and 6) that is used by both importance sampling and beam search.
> 3. The coverage-based and tail-splitting variants of beam search proposed specifically for this task.
> 4. The application of importance sampling on the remaining search space leftover by beam search (referred to as the hybrid method in the paper–Section 4, lines 199 to 219).
>
> We would additionally like to stress that, to our knowledge, we are the first to introduce and investigate predictive querying for black-box neural sequence models.
>
> > [W2.1] … the Q5 cannot be easily handled when C(K, n) gets large…
>
> We appreciate the insightful comments. In the paper, we discussed four non-trivial yet desirable queries of interest; however, as you have mentioned this does not capture all of the possible different probabilistic queries that could be posed under our proposed framework. The two main factors influencing the difficulty of estimating these queries, as discussed in the paper, are primarily the entropy of the underlying model and the size of the query space $|\mathcal{Q}|$.
>
> With queries that fall under the Q5 category (e.g., how likely will the event $a$ happen $n$ times in the next $K$ steps? $p_\theta^*( N_{a}(K)=n)$), while in the limiting case C(K,n) will become the dominant scaling factor for the size of the query space, in many circumstances this actually isn’t as impactful as one might expect. For instance, it can be shown that the query space of a Q5 query is larger than that of a hitting time query (Q3) evaluated one step further (at $K+1$) if $\sqrt[n]{C(K,n)} > V$ where $V$ is the vocabulary size. Looking at our main set of experiments from Section 5, we can see that when evaluating hitting time queries up to 11 steps into the future ($K=11$), these queries had a slightly larger query space of than that of $p_\theta^*(N_{a}(K)=n)$ for $K=10, \forall_{0\leq n \leq K}$ and thus can be seen as comparable in complexity. This behavior easily extends to larger values of $K$ (e.g., $K \approx 100$), especially for larger vocabulary sizes. This is due to the $n^\text{th}$ root dramatically slowing down the factorial growth in the inequality shared above.
>
> We greatly appreciate this question and we plan  to add this discussion and related points to the main paper and the Appendix.
>
> > [W2.2] … In fact, only hitting time (Q3) is tested in section 5.
>
> In the main paper our primary focus was on evaluating hitting time queries as queries in both classes Q4 and Q5 can essentially decompose into summations of mutually exclusive hitting time queries. As such, given roughly equivalently-sized search spaces the performance shown for hitting time queries would extend to queries under Q5. That being said, we also did some investigations for queries under Q2 and Q4 which can be seen in Sections F.3 & F.6 and in Figures 12 & 14 in the Appendix.
>
> We will be sure to clearly emphasize this in the main paper.
>
> > [W3.1] … actual metric relies on the sampling …
>
> We thank the reader for making this comment as our experimental results could have been clearer in this regard. One important clarification we would like to make is that we do in fact use exact ground truth for experiments where ground truth is computable, and rely on sampling methods otherwise. We will make this distinction clearer in the final version of the paper.
>
> We found we could reasonably compute absolute ground truth for queries that possessed a path space size $|\mathcal{Q}|$ < 30 million in our experiments. For this set of queries (generally, queries where $K < 5$), we do compute ground truth and leverage it as our reference measurement (lines 266-267). These results are reported in Figure 3 (lines 270-271) along with error metrics computed against pseudo ground truth ($K \geq 5$). We will make this clearer in the paper.
>
> For the instances where computing exact ground truth is not possible,  we instead use a sampled result as a reference point (referred to as pseudo-ground truth). We would like to emphasize that these samples are generated using an _unbiased estimator_ and produced such that they possess, at most, a pre-specified maximum variance to control for error. This maximum variance is chosen such that the margin of error between the absolute ground truth and pseudo ground truth is orders of magnitude smaller than the margin of error between the pseudo ground truth and the various approximations used in experiments.
>
> _Please refer to Part 2/2 for the rest of our replies._

---

> > ### Author Response · Authors · 2022-08-01
> > **Response to Reviewer yqp6 (PART 2/2)**
> >
> > > [W3.2] … No baselines are available, which makes it hard to judge the contribution of this paper. …
> >
> > Our experimental findings described in the main paper (Figure 3) are further augmented in the Appendix with two additional baselines (Section F.2). Namely, we examine the performance of Monte Carlo sampling from a uniform proposal distribution as well as naive model sampling (direct MC sampling of $\mathbb{E}\_{ x\_{1:K} \sim p_\theta^*}[1(x_{1:K} \in \mathcal{Q})]$). These methods were not included in the main paper as their results were highly non-performant, as shown in Figures 10 and 11. We will be sure to mention these additional baselines in our main paper.
> >
> >
> > > [W3.3] … A suggestion would be training a supervised model for each K to predict the hitting time.
> >
> > We thank the reviewer for this suggestion. We believe that there may be two possible interpretations to this suggestion. As such, we will list out and respond to each interpretation below.
> >
> > _Interpretation 1: Train the model using real data._
> >
> > While tempting, this approach has one main drawback that prevents this from being useful as a baseline. Assuming we could train this new model, we still lack a way of reliably evaluating any method (either this model or our proposed techniques applied to an existing autoregressive model) on real data. Many queries that could be asked do occur naturally in real data; however, this approach is likely to possess  a severe class imbalance between a sequence satisfying a query and a sequence not satisfying it. Without this in place, the best we can do is measure how different the beliefs are between this newly trained model and the existing one but we would have no way of knowing which is better.
> >
> > Nonetheless, we agree with the reviewer that  designing a reliable and effective evaluation of querying methods using  real data is a valuable direction to explore and a worthy subject for future research.
> >
> > _Interpretation 2: Train the model using data that has been generated from an already existing autoregressive model $p_\theta$. The new model would be modeling the existing model’s beliefs rather than trying to reflect the reality that generated the original training data._
> >
> > We also think this is worthy for future work as a possible use-case for our proposed query estimation methods; however, in terms of using this as a potential baseline we run into the same issue mentioned in the other interpretation as we would still need to refer to some form of “ground truth” for the original model’s beliefs. As such, we feel the best way to generate a proxy for ground truth would be to use the proposed sampling procedures while controlling for the variance as described earlier in our response.
> >
> > > [Q1] (Effective Sample Size?)
> >
> > We thank the reader for this suggestion! This indeed could be very insightful. We will be sure to incorporate effective sample size into our findings for the final version of our paper.
> >
> > > [Q2] (Importance Sampling Resampling?)
> >
> > We thank the reviewer for pointing out the potential applicability of ISR to our problem! This method definitely seems applicable and beneficial for our approach. From what we understand, this could be readily applied in instances when using importance sampling as outlined in our paper without any special considerations. This could be beneficial in that it could further improve the variance of the produced estimates. While we believe that our current set of experiments are sufficient to demonstrate the value of our novel proposal distribution in a variety of use cases, we will be sure to add discussion in the paper on how this could be further potentially improved through the addition of ISR.
> >
> > > [Q3] (Balance of IS and BS in Hybrid)
> >
> > We thank the reviewer for this insightful question and comment! The hybrid method we present in our paper balances between beam search and importance sampling with tail-splitting beam search. Tail-splitting beam search dynamically selects the number of beams at each search step with the heuristic detailed on lines 214-219. This is particularly suitable in low-resource settings as it can be interpreted as discovering the minimum number of beams needed to maximize the increase in variance reduction in the path space. In practice, this usually minimizes the amount of computation applied to beam search, leaving a general computation ratio of anywhere from 2:1 to 6:1 between importance sampling and beam search.
> >
> > It is also worth noting that we do not provide a  theoretical justification for our tail-splitting heuristic. It is quite possible that alternative and more effective  beam-selection heuristics could be derived to suit particular  applications. For example, we posit that allocating additional computation to beam search in the hybrid method may be preferable when over-estimating a query is particularly undesirable.  The flexibility of this heuristic could be viewed as a strength of our proposed methodology.

---

> > > ### Comment · Reviewer_yqp6 · 2022-08-08
> > > **Thanks for the detailed response.**
> > >
> > > Thanks for the detailed response. The authors should consider adding their comments to the final version of the paper, especially other baselines. I raise my score to 6.

---

### Official Review · Reviewer_6x57 · 2022-07-12

**Rating:** 7
**Confidence:** 3
**Soundness:** 3 good
**Presentation:** 4 excellent
**Contribution:** 3 good

**Summary:**

The paper aims at solving a task called predictive querying, which aims at producing the probability that probabilistic queries such as “when will event A occur next” or “what is the probability of A occurring before B,” happens in an autoregressive (RNN/Transformer) model. The paper categorizes the probabilistic queries into five categories, and in principle, most of them are intractable with a nearly O(V^K) complexity, where V is the vocabulary size and K is the step size. The paper proposes two approaches to estimate the predictive querying, one is importance sampling, and the other is approximation within beam search. On empirical evaluation, a hybrid of these two techniques works the best on three user behavior and two language sequence datasets.


**Questions:**

1. Given the five probabilistic query categories, the paper only investigates the hitting time queries. Is the proposed method able to generalize to the other categories?


**Limitations:**

Yes

**Strengths And Weaknesses:**

Strengths

1. The paper is the first work that studies predictive querying on modern autoregressive models, and provides effective estimation techniques including importance sampling and approximation in beam search to estimate the probability of these queries, with proven convergence.

2. On several datasets across different domains, the paper shows that the proposed hybrid method achieves the best relative errors compared to the baseline methods.

3. The paper shows that in general, the hybrid method is the best, but in lower entropy regimes such as Mobile Apps, the beam search approximation can outperform the importance sampling and hybrid method. Such a result is informative and may inspire further research.

Weaknesses

1. It is a bit unclear to me what queries and data the paper actually uses in the experiments. It would be better to demonstrate with a few text examples for each dataset.

---

> ### Author Response · Authors · 2022-08-01
> **Response to Reviewer 6x57**
>
> We thank the reviewer for their time, insightful comments, and questions. We have provided our responses below.
>
> > [W1] It is a bit unclear to me what queries and data the paper actually uses in the experiments. It would be better to demonstrate with a few text examples for each dataset.
>
> We agree that we could have explained this more clearly. For the main experiments presented in the main paper (see Section 5), all of the results shown (e.g., Figures 4 and 5) are aggregated from data that was collected using the same procedure. Below we explain this in detail and we will add this level of explanation in the final version of the paper (in either the main paper or the Appendix depending on space).
>
>
> 1. Sample a sequence $x$ of length $n$ from the test set.
> 2. Using a model $p_\theta$ that was trained on the training split of the dataset, condition on the first 5 elements, $x_{1:5}$.
> 3. Then, using the proposed method of interest (e.g., importance sampling, coverage-based beam search, etc.), approximate the query of interest for the future continuation of the sequence–typically K steps into the future, $x_{6:6+K}$. As mentioned in lines 256 and 257 of the main paper, the main experiments pertain to hitting time queries $\tau(a)=K$ where $K=3,\dots,11$ and $a$ is determined as being equal to the actual value of $x_{6+K}$ from the sampled sequence. We ensure $6+K\leq n$.
> 4. Repeat steps 1-3 over 1000 sequences (unless the dataset is WikiText, in which case do 100 sequences).
> 5. Against either $p$ determined by absolute ground truth, or pseudo-ground truth $p$ computed via sampling, compute relative absolute error (RAE) of the form $\frac{|\hat{p} - p|}{p}$. For a given dataset, query estimation method, and budget, compute the median RAE over all 1000 (or 100 in the case of WikiTest) sampled sequences, where a sampled sequence is a history-target tuple $[\mathcal{H}, a]$. Examples can be found in Figures 3-4.
>
> Additionally, we appreciate the suggestion for providing explicit examples of some of these queries. Below are 2 example queries:
>
> **Amazon Reviews**:
> - **History** $\mathcal{H} = [ \texttt{log on}, \texttt{ click}, \texttt{ add to cart} ]$
> - **Query Class**: “A” before “B” query (Q4)
> - **Formalism**: $p_\theta^*(\tau(A) < \tau(B)) = p_\theta^*(\texttt{purchase} < \texttt{log off})$
> - **Description**: Given a user’s online engagement history, what is the probability that they purchase something before they navigate away from the page?
>
> **Shakespeare**:
> - **History** $\mathcal{H} = [ \text{“t”}, \text{“h”}, \text{“o”}, \text{“u”}, \text{“<space>”} ]$
> - **Query Class**: Combination query (Q5)
> - **Formalism**: $p_\theta^*(N_{\texttt{vowel}}(10)=4)$
> - **Description**: In the next 10 characters, what is the probability that 4 of them are vowels?
>
> These examples, and more, will be added to the Appendix.
>
> > [Q1] Given the five probabilistic query categories, the paper only investigates the hitting time queries. Is the proposed method able to generalize to the other categories?
>
> While the main-paper experiments focus on the hitting time queries, in the Appendix we have also provided investigations pertaining to marginal token distributions $K$ steps in the future, $p_\theta^*(X_K)$, as well as  general “A” before “B” style queries, $p_\theta^*(\tau(A) < \tau(B))$. For the former, please refer to Section F.3 and Figure 12 in the Appendix. For the latter, please refer to Section F.6 and Figure 14 in the Appendix.
>
> Also, we would like to emphasize that  our proposed methods are able to generalize to any of the mentioned categories in the paper, as well as many others not discussed. So long as a query can be represented in our proposed framework (see Section 3 and Equations 1-3) then our proposed methods are applicable.
>
> Moreover, we mention in the main paper (lines 257-259) and elaborate in Appendix section F.6 (lines 307-316) that our focus on hitting-time queries is motivated by the fact that more advanced queries (e.g. Q4, Q5) can be reduced to operations on individual hitting-time queries. Specifically, we can see below the deconstruction of Q4 (“a” before “b”) into hitting time operations:
>
> $$
> p_\theta^*(\tau(A) < \tau(B) ) = \sum_{k=1}^\infty p_\theta^*(\tau(A)=k, \tau(B)>k) = \sum_{k=1}^\infty \sum_{a\in A}p_\theta^*(X_k=a, X_{<k}\in (\mathbb{V}\setminus(A\cup B))^{k-1})
> $$
>
> In words, the probability that an element “a” occurs before element “b” decomposes into the sum over hitting time queries that determine the probability that the $X_K = a$ and that element “b” has not occurred at all during $X_{1:K-1}$, summed for all values $K=1 \rightarrow \infty$ (or some large value of $K$ in the finite case).
>
> We appreciate this being pointed out by the reviewer and will make sure to make these points clearer in the final version of the paper (and to provide additional detailed explanations in the Appendix), given that our experimental focus on hitting time queries is an essential aspect of our paper.

---

> > ### Comment · Reviewer_6x57 · 2022-08-08
> > **Thanks for the clarification**
> >
> > I thank the authors for the clarification.
> >
> > W1. Thanks for the detailed explanations and for showing the examples queries. I believe this would increase the readability of the paper.
> >
> > Q1. Thanks for the additional results in the appendix.
> >
> > Since my concerns are solved by the author response, I am happy to increase my score to 7.

---

### Meta-Review · Area_Chair_Jpy7 · 2022-08-25

**Recommendation:** Accept
**Confidence:** Certain

**Metareview:**

The paper tackles an important question: how to answer probabilistic queries about future steps, such as “when will event A occur next”, in the context of autoregressive neural models that are widely used in many applications nowadays. While similar questions have been studied in stochastic processes, it has not received much attention in inference on autoregressive models. This paper formulates the problem, provides a framework, develops baselines, and proposes improved method based on importance sampling and beam search. All reviewers agree that the paper opens an interesting problem space and is technically solid. Therefore, I recommend acceptance.

**Award:**

No

---

### Decision · Program_Chairs · 2022-09-14

Accept